# The observed influence of local anthropogenic pollution on northern Alaskan cloud properties

Maximilian Maahn[1,2], Gijs de Boer[1,2], Jessie M. Creamean[1,2], Graham Feingold[2], Greg M. McFarquhar[3], Wei Wu[4,5], and Fan Mei[6]

[1]University of Colorado, Cooperative Institute for Research in Environmental Sciences, Boulder, Colorado, USA
[2]National Oceanographic and Atmospheric Administration, Earth System Research Laboratory, Boulder, Colorado, USA
[3]University of Oklahoma, Cooperative Institute for Mesoscale Meteorological Studies, Norman, Oklahoma, USA
[4]University of Illinois at Urbana-Champaign, Urbana, Illinois, USA
[5]National Center for Atmospheric Research, Boulder, Colorado, USA
[6]Pacific Northwest National Laboratory, Richland, Washington, USA

*Correspondence to:* Maximilian Maahn (maximilian.maahn@colorado.edu)

**Abstract.**

Due to their importance for the radiation budget, liquid-containing clouds are a key component of the Arctic climate system. Depending on season, they can cool or warm the near-surface air. The radiative properties of these clouds depend strongly on cloud drop sizes, which are governed in part by the availability of cloud condensation nuclei. Here, we investigate how cloud drop sizes are modified in the presence of local emissions from industrial facilities at the North Slope of Alaska. For this, we use aircraft in-situ observations of clouds and aerosols from the 5th Department of Energy Atmospheric Radiation Measurement (DOE ARM) Program's Airborne Carbon Measurements (ACME-V) campaign obtained in Summer 2015. Comparison of observations from an area with petroleum extraction facilities (Oliktok Point) with data from a reference area relatively free of anthropogenic sources (Utqiaġvik/Barrow) represents an opportunity to quantify the impact of local industrial emissions on cloud properties. In the presence of local industrial emissions, the mean effective radii of cloud droplets are reduced from 12.2 to 9.4 $\mu$m, which leads to suppressed drizzle production and precipitation. At the same time, concentrations of refractory black carbon and condensation nuclei are enhanced below the clouds. These results demonstrate that the effects of anthropogenic pollution on local climate need to be considered when planning Arctic industrial infrastructure in a warming environment.

## 1 Introduction

Liquid-containing clouds are a significant modulator of the Arctic climate system's radiation budget. Their properties impact both shortwave and longwave radiative transfer, resulting in seasonally-dependent influences that include both net cooling and warming of the Arctic surface (Intrieri et al., 2002; Shupe and Intrieri, 2004), and various forms of cloud feedbacks (Colman, 2003). At the same time, liquid cloud droplet number concentration and size are influenced by the number of available cloud condensation nuclei. It has been proposed that this has an effect on cloud albedo, life cycle and longwave emissivity (Twomey, 1976; Albrecht, 1989; Garrett and Zhao, 2006). Long range transport of aerosol particles from lower latitudes in winter and early spring (Arctic haze) and episodic forest fires in summer can lead to higher aerosol concentrations (Shaw,

1995; Law and Stohl, 2007), which have been found to modify liquid and mixed phase cloud properties (Garrett et al., 2004; McFarquhar et al., 2011; Jackson et al., 2012; Zamora et al., 2016). Besides these transported emissions, the Arctic is an environment that is generally relatively clean with respect to anthropogenic emissions (Quinn et al., 2002, 2009). The generally low cloud condensation nuclei (CCN) concentrations make clouds particularly susceptible to an increase in CCN concentration

(Platnick and Twomey, 1994). For example, Croft et al. (2016) showed that emissions from seabird-colonies can significantly modify radiative properties of Arctic summertime cloud. In comparison to other regions, there are only few sources of local anthropogenic emissions north of the Arctic Circle, which are mainly related to ship traffic and petroleum as well as natural gas extraction facilities (Law and Stohl, 2007). While emissions from ships are expected to rise due to the retreating sea ice, emissions from resource extraction are expected to remain at present day levels (Peters et al., 2011) with an estimated 13% of

the world's untapped oil resources located in the Arctic (Gautier et al., 2009). Local emissions by Arctic petroleum and natural gas extraction facilities have been observed and quantified by aircraft campaigns (Brock et al., 2011; Roiger et al., 2015). These emissions are mostly associated with flaring, but also by regular internal combustion engines. Ødemark et al. (2012) found that black carbon (BC), which is particularly created by flaring (Stohl et al., 2013), results in a modeled positive net radiative forcing of petroleum and natural gas extraction, mainly due to deposition of BC on the snow. Kolesar et al. (2017) showed that

emissions from the Prudhoe Bay area result in in-situ particle growth events in Utqiaġvik (formerly known as Barrow), located around 300 km west of the Prudhoe Bay region. Although these previous studies have demonstrated the potential impact from industrial activities in the Arctic, in-situ aerosol and cloud observations have not been combined in order to study local sources of emissions.

In this work, we show how cloud properties are altered by aerosol particles originating from local anthropogenic pollution

from industrial activities in the Prudhoe Bay area in northern Alaska (Fig. 1), and investigate the influence on processes impacting the cloud life cycle. Even though the work is limited to observations from the North Slope of Alaska, the results are broadly applicable to other Arctic regions with significant industrial activities (e.g. Siberia), although exact details of the types of aerosol effects will be influenced by aerosol concentration, size, and composition. Because of their importance in regulating the surface and top-of-atmosphere energy budgets, we focus here on liquid clouds. Increased cloud droplet concentrations in the

Prudhoe Bay, Alaska area were previously reported by Hobbs and Rangno (1998) although that study could not directly connect these increased concentrations to locally produced aerosol particles due to a lack of aerosol measurements. In this study, we fill this gap by using airborne cloud property and aerosol observations obtained during the US Department of Energy Atmospheric Radiation Measurement (DOE ARM) program's 5th ARM Airborne Carbon Measurements (ACME-V) campaign to study the influence of local pollution on Arctic liquid clouds. An enhanced understanding of the influence is crucial to evaluate the role

of clouds and aerosols in changing Arctic which is warming faster than other regions (Jeffries et al., 2013).

In Section 2 we provide background information on the ACME-V campaigns along with details on the various data sets used to conduct our analysis. Following this, we analyse observed aerosol particle (Section 3) and cloud (Section 4) properties, before combining these to evaluate the interactions between locally-produced aerosols and clouds in Section 5. This evaluation is carried further in Section 6 where we attempt to quantify observed aerosol-cloud interactions. Finally, we provide a summary

and concluding remarks in Section 7.

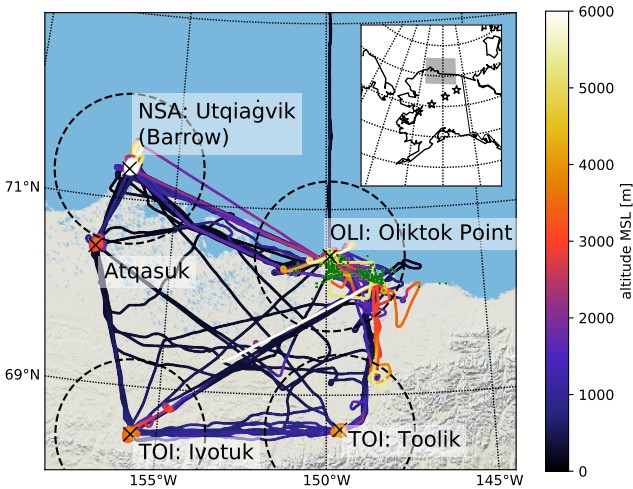

**Figure 1.** Overview of all flights of the ACME-V campaign. Color shows altitude m MSL. The dashed circles indicate 90 km radii around the sites (black crosses), the green dots indicate active oil wells (Data obtained from http://doa.alaska.gov/ogc/publicdb.html in March 2017). The grey inlet shows the location of the region in Alaska and the five assumed sources for forest fire emissions (stars) based on MODIS thermal anomaly observations.

## 2   Data set

The ACME-V aircraft campaign took place from June 1st to September 15th 2015 (Biraud, 2016; ARM, 2016) and consisted of 38 research flights of the ARM Gulfstream G-159 (G-1) aircraft of the ARM aerial facility (Schmid et al., 2014, 2016). Since the campaign targeted trace gas measurements from local and regional sources, a majority of the flight time was spent

5   below 200 m above mean sea level (MSL). However, spirals up to an altitude of 6,000 m were flown in the vicinity of two ARM surface observatories in northern Alaska, Utqiaġvik (formerly known as Barrow or ARM's North Slope of Alaska site, NSA, 71.323°N, 156.616°W) and Oliktok Point (OLI, 70.495°N, 149.886°W). Additional spirals were flown at Toolik (68.628°N, 149.598°W), Ivotuk (68.483°N, 155.754°W), and Atqasuk (70.467°N, 157.436°W) in order to characterise cloud and aerosol properties (Fig. 1). In this work, we compare data within 90 km of OLI and NSA. These two sites form an ideal opportunity

10  to study the effects of local emissions on cloud properties: While OLI is surrounded by industrial activities related to oil and natural gas extraction (with the majority closer than 90 km), no substantial local sources exist in the vicinity of NSA and previous studies have shown only limited advection ($8 \pm 2\%$) of air masses passing through the Prudhoe Bay area to NSA (Kolesar et al., 2017). Despite substantial differences in aerosol properties, the two coastal sites lie only 250 km apart, resulting in very similar synoptic scale forcing, as can be seen from the high correlation between both sites for pressure, temperature,

15  humidity, and wind (Fig. 2). For both sites, north-easterly to easterly winds prevailed during ACME-V (see data set ARM, 1993, updated daily). Additionally, we grouped observations closer than 90 km to the two more continental sites Toolik and Ivotuk into a third data set (labelled TOI). During ACME-V, 156 (60%) of the 258 vertically sampled clouds were classified as liquid (see below for thresholds), showing that liquid clouds are frequent during the summer time in Northern Alaska. Data

obtained during take off and landing have been removed to avoid skewing the comparison by sampling aerosols and clouds at much lower altitudes than elsewhere.

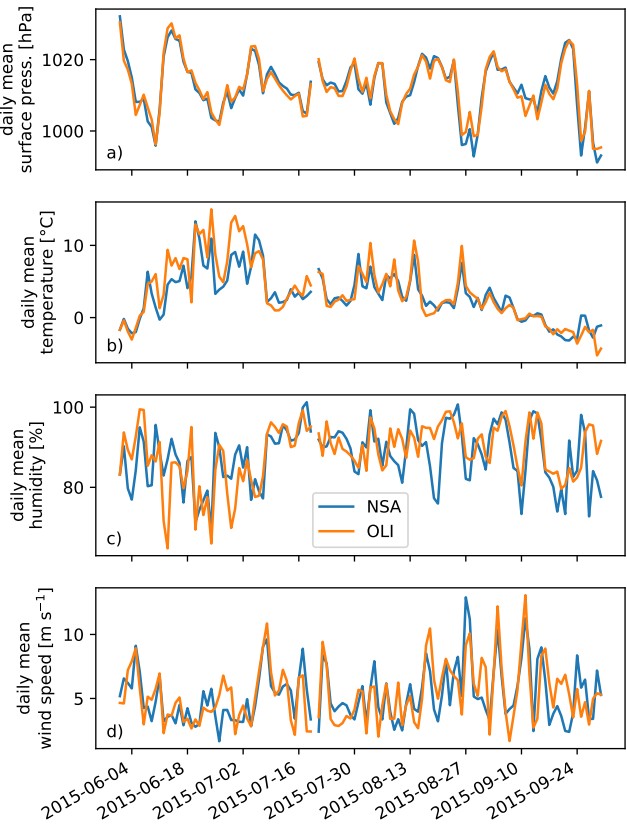

**Figure 2.** Comparison of daily mean values for a) surface pressure, b) 2 m temperature, c) 2 m humidity, and d) 10 m wind speed.

Cloud properties were observed using a combination of forward scattering and optical array probes. The particle size distributions were measured using the forward scattering Cloud Droplet Probe (CDP) manufactured by Droplet Measurement
5  Technologies (DMT), Inc., the Fast Cloud Droplet Probe (FCDP) from Stratton Park Engineering Company (SPEC), Inc., the Two Dimensional Stereo optical array probe (OAP) (2DS, Lawson et al., 2006), and the High Volume Precipitation Spectrometer (HVPS, Lawson et al., 1993) from SPEC, Inc. For the 2DS probe, the evaluation of particle sizing and sample area determination was done following Korolev et al. (1991). The sample areas of CDP and FCDP were determined by their manufacturer using the technique described by Lance et al. (2010). The droplet size response for CDP and FCDP was calibrated
10  weekly using glass beads in field. In addition, LWC was measured by a multi-element water content system (WCM-200) and used for evaluating the in-flight performance of the 2DS, CDP and FCDP (King et al., 1978, 1981, 1985). The raw OAP datasets were processed by the University of Illinois Optical Array Probe Processing Software (Wu and McFarquhar, 2016). In order to merge the cloud droplet size distributions, FCDP data were used for particles less than 50 $\mu$m size, the 2DS was used for particles between 50 and 605 $\mu$m, and the HVPS was used for all particles exceeding 605 $\mu$m. In this study, particle

maximum dimension is used in general to describe the size of cloud and aerosol particles. Liquid clouds were required to have at least 10 cm$^{-3}$ droplets (Lance et al., 2011). We also evaluated the use of a lower cloud threshold (5 cm$^{-3}$ in accordance with Hobbs and Rangno (1998)), but this only increased the number of observed clouds by two, both of which were classified as impacted by forest fire (see below). In order to avoid sampling errors due to small sample sizes, we use the larger threshold

(10 cm$^{-3}$) in our study. In order to remove ice clouds from the data set, the Holroyd habit classification was applied to 2DS and HVPS observations with 1 s temporal resolution, which classifies particles mainly based on a fine detail ratio $F = pd/a$, where $p$ is perimeter, $d$ is diameter and $a$ is area (Holroyd, 1987; Wu and McFarquhar, 2016). The habit classification scheme differentiates between spherical particles, tiny particles which are too small to be classified and various forms of ice crystals. Spherical particles were assumed to be liquid. Tiny particles appear only at the lower end of the 2DS ($< 105\ \mu$m) and HVPS

($< 1125\ \mu$m) size range. They were classified as ice only if other size ranges were not dominated by spherical particles. Otherwise, tiny particles were assumed to be liquid. Data points with more than 100 m$^{-3}$ particles larger than 400 $\mu$m (Lance et al., 2011) classified as ice were removed from the data set. This ensures that observations of spherical ice particles falsely classified as liquid, which likely occur together with larger, more complex shaped ice particles, were removed from the data set as well. Liquid water content (LWC) was obtained by integrating the merged droplet size distribution (DSD), because direct

observations of LWC from the King probe (King et al., 1978) are affected by a decreasing sampling efficiency for (drizzle) drops greater than 30 $\mu$m diameter. Clouds that were observed for less than ten continuous seconds were discarded, while gaps of up to 5 s were permitted once in cloud. Considering the typical true airspeed of the G-1 of 95 m/s, 10 s and 5 s correspond to 950 m and 475 m, respectively, when flying in a straight line. Additionally, only vertically sampled clouds (i.e. the aircraft was constantly ascending or descending) with a sampled vertical extent of at least 20 m were included in this evaluation to allow

for comparison of in-cloud microphysical observations with below-cloud aerosol properties in sections 5 and 6. Therefore, very thin and/or small clouds might be discarded inadvertently. To make the detection of cloud boundaries more robust, the cloud probe data were smoothed using a 10 s running average. Except for the detection of the cloud boundaries, effects of the smoothing are negligible for the presented analysis.

Aerosol particles were sampled through an isokinetic inlet with an upper size cut of 5 $\mu$m (Zaveri et al., 2010; Dolgos

and Martins, 2014). Aerosols in the size range 100 nm to 3 $\mu$m were observed with the Passive Cavity Aerosol Spectrometer (PCASP model 100X, DMT Inc.) covering most accumulation mode aerosols (Colbeck and Lazaridis, 2014). We expect particles measured by the PCASP to be mostly dry, because it was operated with an anti-ice heater. Kassianov et al. (2015) showed for the very same aircraft that this assumption leads to good agreement between calculated (using, among others, PCASP observations) and measured scattering properties. The PCASP was calibrated using both size selected ammonium sulphate particles

and monodisperse polystyrene latex (PSL) spheres. The sizing accuracy was checked weekly in the field using PSL particles following Cai et al. (2013). Unfortunately, another aerosol sampler (Ultra-High Sensitivity Aerosol Sizer, UHSAS), which is able to detect aerosols below the PCASP detection threshold of 100 nm, did not operate during the majority of the ACME-V flights. Two Condensation Particle Counters (CPC, TSI, Inc. models 3025 and 3010) were used to observe total number concentrations of condensation nuclei (CN) for the size ranges 3 nm - 3 $\mu$m and 10 nm - 3 $\mu$m, respectively. CPC calibration

activities included verifying inlet flow rate with a low pressure-drop bubble flow meter, and determining the size-dependent

particle counting efficiency, according to methods defined in Hermann et al. (2007) and Mordas et al. (2008). Unless otherwise stated, only the CPC 3025 featuring a size range of 3 nm - 3 $\mu$m is used in this evaluation. The mass and core size of black carbon (BC), which results from incomplete combustion of biomass and fossil fuels (Schwarz et al., 2008; Bond et al., 2013; Lack et al., 2014), was measured with the Single Particle Soot Photometer (SP2, from DMT Inc.), via incandescence. Thus,
only refractory black carbon (rBC) is observed by the instrument. The applied SP2 calibration methods using ambient BC and fullerene soot are described in detail by Gysel et al. (2011) and Irwin et al. (2013). The fullerene soot and PSL calibration were performed twice during this field campaign and the sensitivity of the SP2 was found to be stable to around 10% for fullerene soot particles resulting in an estimated SP2 measurement uncertainty of 10%. Concentrations of carbon monoxide (CO) were detected with a Los Gatos Research CO/$N_2$O/$H_2$O Analyzer. A counter for cloud condensation nuclei (CCN) was not deployed
during ACME-V. The temporal resolution of the aerosol probes is 1 s with the exception of the SP2 (10 s).

Transported emissions from forest fires can contribute significantly to summertime aerosol loading in the Arctic (Law and Stohl, 2007; Creamean et al., 2017). Therefore, we manually inspected the vertical profiles of rBC and CO, which together can be used to trace biomass burning in otherwise clean environments (Warneke et al., 2009, 2010; Zamora et al., 2016). Typically, these layers are found aloft (Roiger et al., 2015), allowing us to use vertical profiles obtained by the aircraft to aid
in their identification. For each spiral obtained at the two sites, elevated layers with CO $\geq$ 0.1 ppmv or rBC $\geq$ 20 ng kg$^{-1}$ were flagged as potentially associated with forest fires. Local emissions, on the other hand, are expected to be concentrated in the boundary layer. Note that the data impacted by forest fires were only removed for spirals above OLI, NSA, and TOI. For clear-air observations during level flight legs between sites, it is generally impossible to determine whether a layer is connected to the surface or elevated. For ACME-V, Creamean et al. (2017) classified only four flights as impacted by long range transport
from lower latitudes not related to forest fires. During these flights, only a single cloud was sampled in the vicinity of OLI or NSA which had one of the lowest aerosol concentrations measured in the whole data set. Therefore we are confident that our analysis is not strongly impacted by this kind of long range transport events.

The manual inspection was supported by aerosol dispersion simulations executed using version 4 of the Hybrid Single Particle Lagrangian Integrated Trajectory (HYSPLIT) model (Stein et al., 2015). These simulations were forced using 1° data from
the NOAA/NCEP Global Data Assimilation System (GDAS) (Kalnay et al., 1996). Five locations were included as sources (see Fig 1: (1) 62.096°N, 163.632°W, (2) 63.843°N, 159.046°W, (3) 65.294°N, 154.386°W, (4) 66.631°N, 149.023°W, and (5) 67.631°N, 144.087°W). These sources were toggled on or off on a daily basis in correspondence to thermal anomaly observations in the corresponding region (see Fig. 4 of Creamean et al., 2017) from the Moderate Resolution Imaging Spectroradiometer (MODIS) on the Aqua and Terra satellites obtained using brightness temperature measurements in the 4 and 11
$\mu$m channels (Giglio et al., 2003; Giglio, 2013). From each fire location, particle mass concentrations were simulated for 72 h at 100-m intervals from 0 to 5,000 m above ground level (m AGL). Both dry and wet deposition were considered for particles using the default HYSPLIT parametrisations (particle density 6 g cm$^{-3}$, shape factor 1.0). The particle diameter of 0.2 $\mu$m used for the simulations is based on previous observations from fossil fuel and biomass burning sources (Brock et al., 2011; Eck et al., 1999; Rissler et al., 2006; Sakamoto et al., 2015). A dry deposition velocity of 1 x 10$^{-4}$ m s$^{-1}$ was assumed according
to Warneck (1999) while 4 x 10$^4$ L L$^{-1}$ and 5 x 10$^{-6}$ s$^{-1}$ were used to account for in-cloud and below-cloud wet deposition

scavenging, respectively. Radioactive decay and pollutant resuspension were not considered. For the spirals, data identified as originating from forest fire either from manual inspection or according to HYSPLIT, were removed from subsequent analysis. With this approach, we likely removed more clouds from the analysis than required. This, however, ensures that the analysis of the remaining clouds is not biased by influences from forest fires.

## 3   Aerosol properties

The spatial distribution of aerosol observations below 500 m MSL are presented for the CPC, the SP2, and the PCASP in Fig. 3. As discussed above, removal of data potentially impacted by forest fires is only possible for the spirals. Therefore, the data presented in Fig. 3 are limited to observations obtained below 500 m, because transported emissions of forest fires were typically at higher altitudes during ACME-V (for a detailed study of aerosol properties during ACME-V see Creamean et al., 2017). Furthermore, aerosol data flagged as sampled in cloud using the thresholds described in the previous section were discarded in the analysis of aerosol properties due to concerns of contamination of the statistics by shattering of cloud droplets.

A clear local maximum of rBC mass concentration is visible east of OLI in the SP2 data within the 90 km radius (Fig. 3.a) where most petroleum and gas extraction facilities are located (Fig. 1). A comparison of the distributions measured within a 90 km radius around the facilities at NSA and OLI reveals that the median rBC concentration is the same for both regions (4 ng kg$^{-1}$). The tail and the number of outliers of the distributions towards larger concentrations, however, are greater at OLI (90th and 99.9th percentile 17 ng kg$^{-1}$ and 198 ng kg$^{-1}$, respectively) than at NSA (15 ng kg$^{-1}$ and 42 ng kg$^{-1}$, respectively). rBC is a tracer for combustion emissions (Bond et al., 2004). Because the height threshold of 500 m reduces the impact of forest fires, this enhancement is most likely connected to local emissions. CN measurements from the CPC show a spatial pattern similar to the SP2 even though the increased values are distributed over a larger area (Fig. 3.b). For both instruments, the distributions within the 90 km circle belonging to each site are skewed towards higher concentrations and the distributions of both sites are significantly different (1% confidence interval) according to the two sample Kolmogorov–Smirnov (KS) test (Massey, 1951). Further, the difference between the two CPC instruments, which equates to the concentration of CN between 3 and 10 nm diameter, is enhanced in the OLI region and the distribution is significantly (KS-test) different to the one at NSA (Fig. 3.c). Because this quantity is stemming from the difference in two instruments at the limit of their measurement range, the data is used here only in a qualitative way. Freshly emitted soot has been shown to be larger than 15 nm (Zhang et al., 2008), so particles in the 3 to 10 nm size range are likely due to in situ nucleation of aerosol particles from gas phase precursors (i.e., formation of new particles as compared to secondary aerosol formation, where gases condense onto preexisting aerosol, Kulmala et al., 2012). Nucleated aerosols typically have sizes below 3 nm, but quickly grow via condensation and coagulation to sizes > 3 nm (Colbeck and Lazaridis, 2014). This source of nucleated aerosol particles from petroleum and gas extraction activities (e.g., flaring and venting of gas) has been reported by Kolesar et al. (2017) for emissions transported from OLI to NSA.

Unfortunately, we cannot analyse this aerosol nucleation process in more depth given limitations with the instrumentation operated during ACME-V. rBC background concentrations appear to be similar to background observations made by Zamora

et al. (2016) and Roiger et al. (2015). It should be noted that emissions related to forest fires led to concentrations as high as 600 to 1000 ng kg$^{-1}$ during ACME-V (mostly at altitudes above 500 m, Creamean et al., 2017) which were also observed in other data sets (Warneke et al., 2009; Schwarz et al., 2010; Zamora et al., 2016). Consequently, the emissions from anthropogenic sources in the OLI region are about a magnitude lower. In contrast to CO concentrations sampled in air masses originating from

forest fires, low altitude CO concentrations in the OLI region were not enhanced relative to background values (Creamean et al., 2017). The differences between CO and rBC concentrations attributed to forest fires and the concentrations measured in the OLI region show that our approach to use CO and rBC to separate observations impacted by forest fires is feasible.

The PCASP, which detects only particles larger than 100 nm, shows no spatial trends in the vicinity of the two sites (Fig. 3.d). The comparison of the distributions around the facilities shows that the number of aerosols observed by the PCASP is on

average actually slightly larger for NSA than for OLI. This is related to the fact that the median of the distribution is 97 cm$^{-3}$ at NSA and 76 cm$^{-3}$ at OLI. Similar to rBC, the tail of the distributions towards larger concentrations is greater at OLI (90th percentile 200 cm$^{-3}$) than at NSA (184 cm$^{-3}$) and the difference in the distributions is significant according to the KS-test with 1% confidence interval. While it is challenging to clarify the precise cause of the increased mean concentration in detail, we speculate it might be related to transported emissions, including those from forest fires, that have not been properly

removed from the data set because they are highly diluted. Transported forest fire aerosols are often larger than the PCASP detection threshold of 100 nm as shown by Kondo et al. (2011) and Sakamoto et al. (2015). An alternative explanation could be the fact that collision-coalescence and precipitation rates are larger at NSA than at OLI (see next Sec. 4.) resulting in more aerosol processing by precipitation (e.g. Feingold et al., 1996). Cloud-based processing leads to a reduction in aerosol concentration and an increase in aerosol size through conglomoration of cloud droplets (and corresponding aerosol particles) in

the drizzle formation stage and subsequent evaporation. Such effects could increase the number of aerosols within the PCASP measurement size range.

## 4   Cloud properties

Here, cloud properties are compared for flights occurring near NSA and near OLI. In order to evaluate a sufficiently large sample, all observations obtained closer than 90 km to NSA, OLI and the two sites comprising TOI are assigned to the

corresponding site (Fig, 1). Also data obtained at altitudes above 500 m is considered, but clouds impacted by forest fires have been removed based on the vertical profile as mentioned above. This limits the number of observations to 996 1 s data points for OLI, 942 for NSA, and 514 for TOI. The data set contains mostly shallow (median cloud depth 107 m) stratus and stratocumulus clouds with a cloud base between 178 m and 5346 m (median of 1498 m).

When comparing 2D histograms of liquid effective radius and liquid water content for OLI and NSA (Fig. 4, a, b), OLI

values are shown to feature smaller r$_{\text{eff}}$ for the same LWC. The effect is most pronounced for LWC > 0.1 g m$^{-3}$ (Leaitch et al., 1992) while distributions for LWC < 0.1 g m$^{-3}$ are more similar. Note that LWC values below 0.01 g m$^{-3}$ are defined as not in-cloud by some studies (e.g., Matsui et al., 2011; Leaitch et al., 2016), but we decided to show the full data set because the in-cloud definition used here (> 10 droplets cm$^{-3}$) can result in LWC as low as 0.001 g m$^{-3}$ and we wanted to make sure

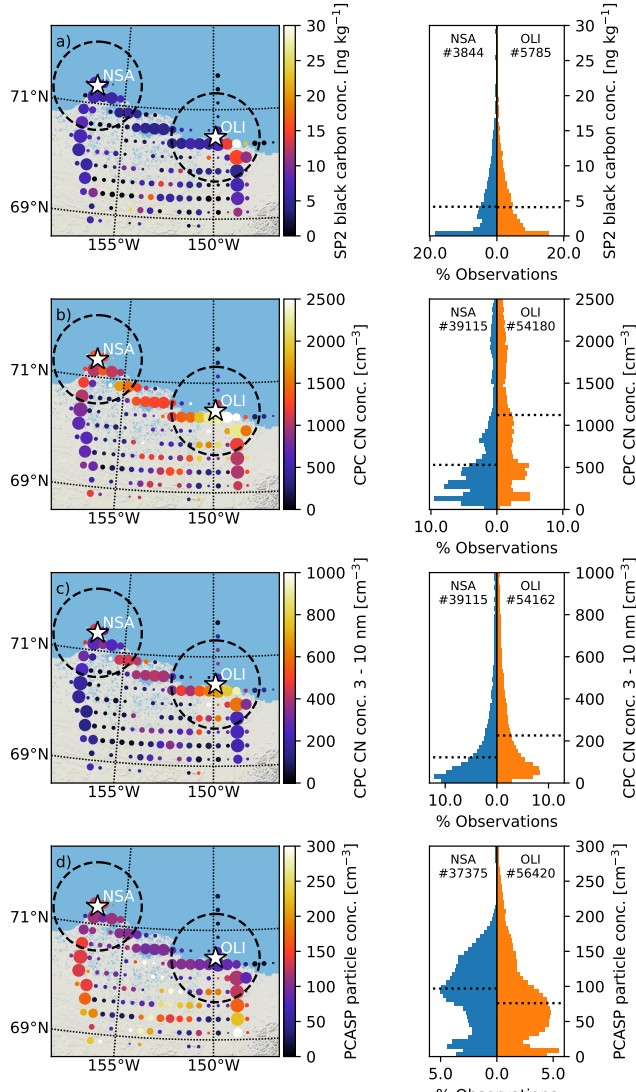

**Figure 3.** Left column: Spatial distribution of mean (a) SP2 refractory black carbon concentration, (b) CPC3025 CN concentrations, (c) difference between CPC3025 and CPC3010 CN concentration corresponding to a size range of 3 to 10 nm, and (d) PCASP aerosol concentration. Only non-cloudy observations below 500 m MSL have been considered. The size of the dots is proportional to the number of observations. The dashed circles correspond to a distance of 90 km. Right column: Here, the distribution of measurements within the 90 km circles are shown, the number above the distribution shows the number of observations. The horizontal dotted bar denotes the median value.

that all cloud data points are included in the analysis. The decrease in $r_{eff}$ supports our hypothesis that CCN concentrations are elevated in the OLI region, since the first aerosol-cloud indirect effect proposes that droplet size is reduced when more CCN are available (all else equal). While droplet $r_{eff}$ observed at NSA cover the full range from droplet nucleation to drizzle (3 to 25 $\mu$m, mean 12.2$\pm$6.9 $\mu$m), $r_{eff}$ values are typically smaller than 16 $\mu$m at OLI (mean 9.4$\pm$4.1 $\mu$m) and observations of

drizzle-sized droplets are rare. The value of 16 $\mu$m is of special interest because it was proposed by Gerber (1996) as a minimal effective radius required to initiate collision-coalescence. Fig. 5.a reveals that this difference is statistically significant for most LWC > 0.1 g m$^{-3}$ according to a Welch's t-test (Welch, 1947) with a 5% confidence interval (Note that a 5% confidence interval is always used in this study unless noted otherwise). For comparison, data obtained in a 90 km radius around Toolik

and Ivotuk (TOI) (Fig. 4, c) reveal that the distribution of observed $r_{\mathrm{eff}}$ at the coastal site in OLI is still larger to the inland sites comprising TOI (mean 7.2$\pm$3.1 $\mu$m) than to the second coastal site NSA. For TOI, the mean $r_{\mathrm{eff}}$ is significantly different than those at NSA for LWC > 0.02 g m$^{-3}$. Fig. 5.b reveals that not only the mean $r_{\mathrm{eff}}$ is reduced at OLI, but also the breadth of the distribution as shown by the standard deviation. This difference is significant for most data points with LWC > 0.1 g m$^{-3}$.

The Albrecht effect proposes that more polluted clouds have longer cloud lifetime due to less efficient collision-coalescence
(Albrecht, 1989). It is not possible to study the cloud life cycle using aircraft in-situ observations, but the potential for impact on cloud life cycle can be estimated by calculating the collection growth rate $C$ (Long, 1974) and precipitation rate $R$. Even though the rate of mass removal from a cloud is an important process impacting cloud life cycle, it is important to note that modifications to $C$ and $R$ cannot be directly translated into modifications in cloud lifetime. This is because a reduction in $R$ could result in a number of feedbacks such as cloud deepening (Stevens and Feingold, 2009) or reduced evaporation just below
cloud base (Feingold and Siebert, 2009) that would act to counter the extending effect of reduced precipitation rate on cloud lifetime.

$C$ describes the mass of drops collected by a unit mass in a unit volume $M$ per time interval $t$. It is the key process for converting cloud drops into precipitation and is estimated by integrating the mass collected by particles with diameter $D_1$ and mass $m_1$ over all size bins:

$$C = \frac{\mathrm{d}M}{\mathrm{d}t} = \int_{D_{\mathrm{min}}}^{D_{\mathrm{max}}} \frac{\mathrm{d}m_1}{\mathrm{d}t} N(D_1)\,\mathrm{d}D_1 \tag{1}$$

where $N(D_1)$ is the particle number distribution function and $D_{\mathrm{min}}$ and $D_{\mathrm{max}}$ are the bounding drop diameters as determined by the cloud probes (0.75 $\mu$m and 8.7 mm). $\frac{\mathrm{d}m_1}{\mathrm{d}t}$ is obtained by integrating the collection kernel $K$ for all smaller size bins (i.e. $D_1 > D_2$) described by the diameter of the collected drops $D_2$

$$\frac{\mathrm{d}m_1}{\mathrm{d}t} = \frac{\pi \rho_w}{6} \lim_{D_1 \to D'} \int_{D_{\mathrm{min}}}^{D'} K(D_1, D_2) N(D_2) D_2^3 \,\mathrm{d}D_2 \tag{2}$$

where $\rho_w$ is the density of liquid water. For simplicity, here we use a simple polynomial approximation of $K$

$$K(D_1, D_2) \approx \begin{cases} 5.78 \times 10^3 (v_1 + v_2) & 20 \leq D_1 \leq 100\mu m \\ 9.44 \times 10^9 (v_1^2 + v_2^2) & D_1 > 100\mu m \end{cases} \tag{3}$$

where $v_i$ is the drop volume corresponding to $D_i$ (Long, 1974; Pruppacher and Klett, 2010). Typical values for $C$ range from $1 \times 10^{-16}$ kg m$^{-3}$s for LWC = 0.001 g m$^{-3}$ to $1 \times 10^{-5}$ kg m$^{-3}$s for LWC = 1 g m$^{-3}$. Note that our approximation does not consider the impact of turbulence and droplet charge on $C$. This might lead to considerable uncertainties, which have—to

the authors' best knowledge—not been fully quantified. Because we are interested how $C$ is modified in the OLI region, we focus on the ratio of $C$ determined at NSA and OLI which should reduce the uncertainty of $C$. Fig. 6 shows the ratio between NSA and OLI of $C$ as a function of $r_{eff}$ and LWC. It can be seen that $C$ is decreased at OLI in comparison to NSA by up to one order of magnitude for constant LWC and $r_{eff}$. This is caused by reduced broadening of the drop size distribution towards large drops at OLI (Fig. 5.b), consistent with the experiments of Gunn and Phillips (1957), who produced similar results when ingesting polluted background air into their cloud chamber. The difference between both sites is significant for most values with sufficient number of observations for both sites (see Fig. 4). However, small absolute increases in $C$ for small $r_{eff}$ are also crucial for triggering the positive feedback of drop growth due to collision-coalescence. When evaluating the potential impact of reduced $C$ on cloud life cycle, one also has to consider that typical $r_{eff}$ values are reduced at OLI in comparison to NSA for the same LWC (Fig. 4, a, b). Therefore, we estimate the mean growth rate $\bar{C}$ averaged over $r_{eff}$ as a function of LWC (Fig. 6, a) red lines). Doing so reveals that, for constant LWC, $\bar{C}$ is reduced by 1 to 1.5 orders of magnitude at OLI. The offset is significant and surprisingly constant for LWC larger than 0.01 g m$^{-3}$. Differences in $C$ also translate to different rain rates $R$, which can be estimated by integrating the measured DSD and applying the fall velocity parametrisation of Khvorostyanov and Curry (2002) which provides a continuous solution over the entire drop size range in dependence of the Best and Reynolds number. Like $C$, $R$ is reduced by up to one order of magnitude for constant LWC and $r_{eff}$ (Fig. 6, b). Averaging over all $r_{eff}$ enhances the effect and leads to differences of up to two orders of magnitude for $R$ as a function of LWC. This effect is statistically significant for LWC $> 0.02$ g m$^{-3}$.

Parameterizations of $C$ and $R$ are crucial in numerical models to transform cloud liquid water into rain droplets and to remove condensate from the atmosphere. Typically, numerical weather and climate models include either one (LWC, one-moment schemes) or two (LWC and drop concentration or $r_{eff}$, two-moment schemes) prognostic variables per hydrometeor species. Our comparison of $C$ and $R$ for both sites reveals, however, that these quantities vary by up to one order of magnitude for constant LWC and $r_{eff}$ (which would be equivalent to a two-moment scheme). Considering only LWC (i.e. one-moment scheme) increases the differences to 1.5 to 2 orders of magnitude. As a consequence, additional moments or the full particle size distribution need to be considered in order to accurately estimate $C$ and $R$ in these models. Otherwise, a parametrisation of $C$ or $R$ relying only on LWC (LWC and $r_{eff}$) might be biased up to 1.5 to 2 (1) orders of magnitude for one-moment (two-moment) schemes.

## 5 Aerosol cloud interaction

So far, we have demonstrated that there are differences in aerosol properties and cloud properties between NSA and OLI. This is in general agreement with the findings of Hobbs and Rangno (1998) who found an increase in droplet number concentration when flying over Prudhoe Bay. In this section, we present evidence that these changes are indeed connected to local industrial activities centred around the Prudhoe Bay oil fields.

In order to evaluate the likelihood that Prudhoe Bay emissions impacted different portions of the ACME-V flights, we use the HYSPLIT dispersion model. Simulations were completed using one continuously emitting source located over the Prudhoe Bay

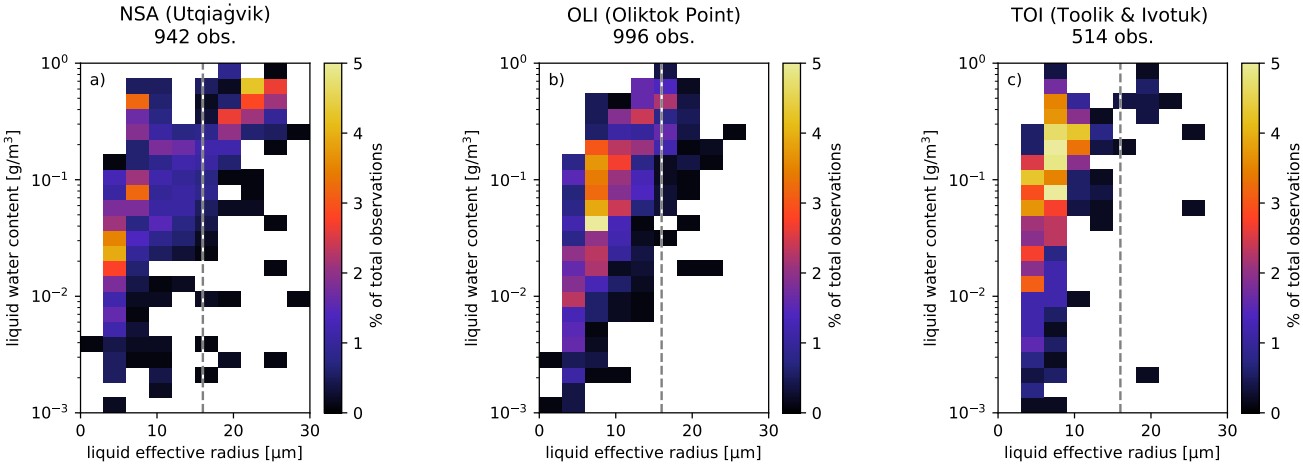

**Figure 4.** 2D-Histogram of number of observations as a function of effective radius $r_{eff}$ and liquid water content LWC for NSA (Utqiaġvik/Barrow) (a), OLI (Oliktok Point) (b) and TOI (Toolik/Ivotuk) (c). The dashed line indicates a $r_{eff}$ value of 16 $\mu$m.

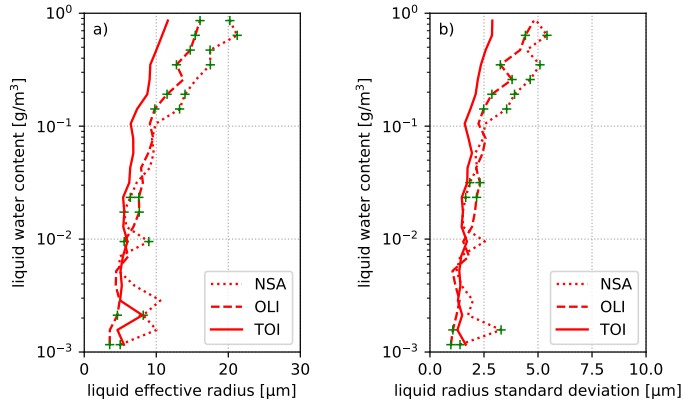

**Figure 5.** a) Mean $r_{eff}$ and b) mean standard deviation as a function of LWC for the data presented in Fig. 4. The green crosses indicate a significant difference between OLI and NSA (5% confidence interval).

oilfields (70.2556°N, 148.3384°W), using a configuration similar to that discussed above for evaluation of wildfire emissions. Note that given the coarse resolution of the forcing model (1°) and the complexity of the Arctic boundary layer, HYSPLIT is used here only in a qualitative way, and not to select locally impacted clouds. For OLI (NSA), 50% (16%) of all data points observed within clouds during ACME-V can be traced back to surface emissions (i.e. mass concentration $> 0$ according to HYSPLIT) originating from the Prudhoe Bay oilfields. The 16% determined for NSA is roughly twice that presented in Kolesar et al. (2017). However, they studied aerosol concentration at the surface instead of aloft and used a multi-year data set, which could introduce substantial variability from the 3-month period evaluated here. The HYSPLIT simulations (Fig. 7) show that the mass concentration originating from local pollution sources can be a substantially higher at OLI than at NSA which is

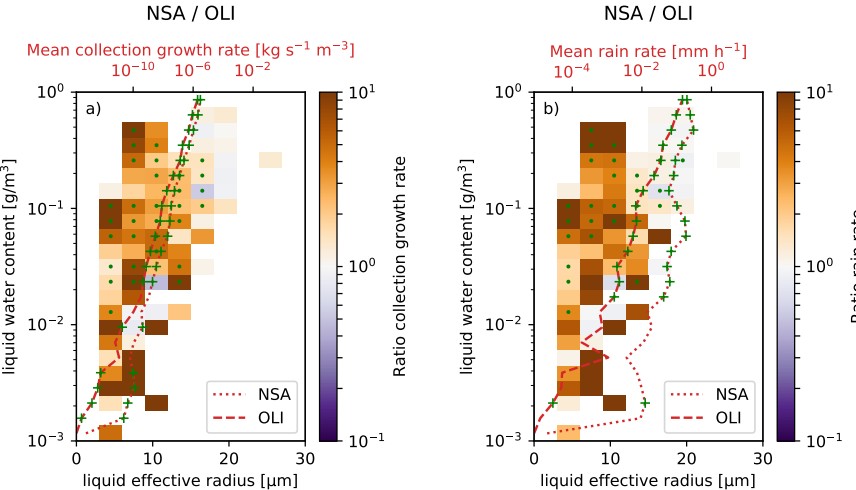

**Figure 6.** As Fig. 4, but with the coloured shading representing the site-to-site ratio (NSA/OLI) of the collection kernel growth rate $C$ (a) and the rainrate $R$ (b). The rates averaged over $r_{\mathrm{eff}}$ are shown in red for OLI (dotted) and NSA (dashed). The green dots and crosses highlight data points with a significant difference (5% confidence interval).

consistent with the observed difference of the tail of the PCASP distribution (Fig. 3.d). These simulations indicate that relative to NSA, the number of clouds impacted by local emission is higher at OLI and these clouds are impacted by a larger amount of aerosol particles by mass. However, an impact of local emissions on cloud properties is also possible at NSA, although less frequently than at OLI. The bin sizes in Fig. 7 were reduced in order to investigate the variability between clouds. This reveals that only a subset of clouds is associated with local pollution according to HYSPLIT. Note that HYSPLIT provides only a relative emission rate in 'mass m$^{-3}$' because the actual emission rate in the Prudhoe Bay region is unknown.

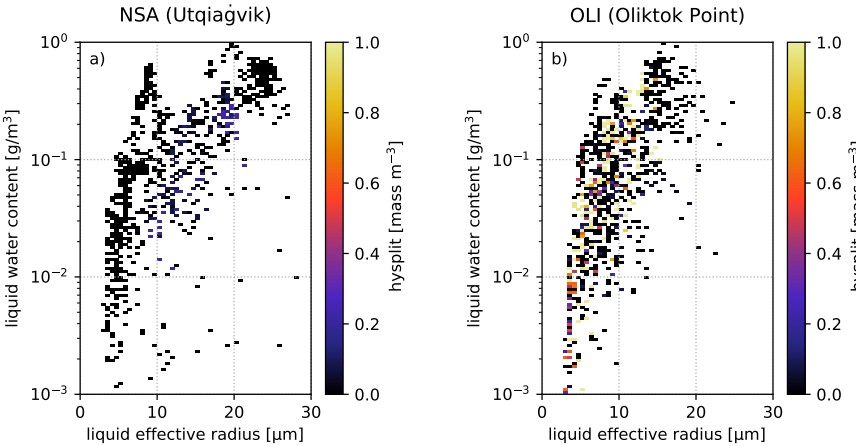

**Figure 7.** As Fig. 4, but with smaller bin size and mass concentration of local emission according to the HYSPLIT model for OLI (a) and NSA (b).

Fig. 8 relates in-cloud observations of LWC and $r_{eff}$ to below-cloud observations of rBC similar to the approach of Jackson et al. (2012). As a consequence, a single below-cloud aerosol value is assigned to every data point within the same cloud based on the assumption that aerosol properties are not changing on the scale of individual cloud profiles. The below-cloud values are averaged, whenever possible, over 30 seconds with a 3 second gap to the cloud base to avoid issues with time synchronisation across instruments or cloud particle contamination of aerosol probe measurements. It shows that the smallest $r_{eff}$ are connected to enhanced rBC concentrations ($> 10$ ng kg$^{-1}$) for both sites. Note that for both sites, these high concentrations correspond in each case to a single cloud. However, for OLI there are more enhanced rBC concentrations ($> 4$ ng kg$^{-1}$) for intermediate $r_{eff}$-values (5-10 $\mu$m) consistent with Fig. 3.a. It is interesting to note that this intermediate region is consistent with enhanced local particle concentrations according to HYSPLIT. rBC can originate from biomass burning as well as anthropogenic sources, but particle size is smaller for the latter (Schwarz et al., 2008). A comparison of rBC core size (Fig. 9) shows that black carbon particles are generally 50 to 300 nm smaller at OLI than at NSA. Together with the collocated enhanced HYSPLIT concentrations, this supports the idea that rBC measurements around OLI are associated with local emissions from Prudhoe Bay and not transported fire emissions. The coincidence of increased rBC concentrations with reduced $r_{eff}$ for OLI might indicate that the observed rBC acted as a CCN. However this would require the rBC to be coated with more hygroscopic material (e.g. sulphate), because pure rBC does not serve as efficient CCN (Weingartner et al., 1997). Note that the SP2 detects the non-coated size of the particles' rBC core, meaning the particles are larger when coated and can potentially act as a CCN despite their small core size.

Similar to Fig. 8, the below-cloud CPC CN concentration is shown in Fig. 10. This figure also indicates an impact of local emissions in the OLI data: CN-observations are enhanced (partly $> 1000$ cm$^{-3}$) at OLI for all $r_{eff}$ even though variability is high (compare also Fig. 3.b). The CN observations are dominated by Aitken mode particles which are typically too small to act as a CCN. This is consistent with the fact that Fig. 10 does not show a correlation between CN concentration and $r_{eff}$. Even though the CN dominating the CPC observations are likely too small to act as CCN, these small particles can grow to accumulation mode quickly given sufficient gaseous precursors, potentially creating a particle population capable of acting as CCN (Jaenicke, 1980).

For the PCASP (Fig. 11), the aerosol concentration is $> 100$ cm$^{-3}$ for small $r_{eff}$ values and $< 20$ cm$^{-3}$ for large large $r_{eff}$ values. Note that for NSA, PCASP data corresponding to some of the largest $r_{eff}$ have been flagged as invalid during quality control and are missing in the figure. The fact that the response of $r_{eff}$ to PCASP aerosol concentrations is—for constant LWC—almost monotonic for both sites is likely because the PCASP covers the aerosol size range most relevant to droplet nucleation and is consistent with the first indirect effect. A different behaviour would indicate that clouds react differently to the same PCASP concentration (which covers most of the accumulation mode size range, see also Sec. 6). However, even though similar PCASP concentrations lead to similar $r_{eff}$ for both sites, differences still exist relating to the breadth and tail of the the distributions, as can be seen from differences in $C$ and $R$ (Fig. 6).

Analysis of the relationship between clouds and HYSPLIT concentrations, rBC and CN shows that some, but not all, clouds at OLI are impacted by local pollution. rBC and CN concentrations are enhanced in the OLI region (Fig. 3) which is probably related to anthropogenic combustion processes and gas flaring/venting, respectively. Therefore we used these quantities as

indicators to isolate clouds impacted by anthropogenic emissions even though there also exist other local sources of small particles (Tunved et al., 2013). Clouds, whose mean below-cloud rBC or CN concentration is above the median concentrations shown in Fig. 3 (4.1 ng kg$^{-1}$ and 1122 cm$^{-3}$, respectively), are identified as potentially impacted. When using this criterion, 10 of 24 (3 of 16) clouds at OLI (NSA) are identified as potentially locally influenced (Fig. 12). For NSA, two of the three clouds corresponded to either extremely low CN ($< 20$ cm$^{-3}$) or rBC ($< 1$ ng kg$^{-1}$) values, making a connection to anthropogenic activities unlikely. But, the clouds classified as anthropogenic at OLI correspond mostly to enhanced concentrations of rBC (Fig. 8) and CN (Fig. 10) and mid-sized r$_{eff}$ (5 - 15 $\mu$m). Fig. 12 shows how the PCASP concentrations of the potential locally affected clouds compares to the clouds classified as affected by forest fires (these clouds are removed in all other Figures except Figs. 12 and 14) and the remaining, non-classified clouds referred to as 'other'. Note that forest fire emissions were also present in the vicinity of NSA, but cloud measurements from these time periods did not pass the quality control measures implemented (continuously ascending or descending profiles). It is striking that the clouds classified as associated with forest fire have a significant (t-test), 6-times larger linear mean PCASP concentration than the clouds classified as locally affected at OLI (510 vs 80 cm$^{-3}$). Despite this big difference, clouds classified as locally affected still feature PCASP concentrations significantly larger than the clouds classified as other (35 cm$^{-3}$). We conclude that CN and rBC particles, which were used to classify local clouds, have the potential to grow to accumulation mode particles measured by the PCASP. For NSA, however, the mean PCASP concentration for clouds classified as other is not significantly (t-test) different from the clouds classified as locally affected at OLI. This is also true when including the three clouds classified as locally affected (from which only one is potentially local as discussed above). This is consistent with the findings of Fig. 3 which shows that the general PCASP concentration background is enhanced at NSA in comparison to OLI.

For the clouds classified as locally affected at OLI, the difference in rBC, CN and PCASP particle concentration above and below the cloud is presented in Fig. 13. This figure confirms that clouds impacted by local emissions feature higher aerosol concentrations below the cloud than above. This also supports our assumption that below-cloud aerosol properties are most relevant for clouds impacted by anthropogenic emissions, which is also true for the remaining anthropogenically influenced cloud at NSA (not shown).

## 6   Quantification of aerosol cloud interaction

Various attempts have been carried out to quantify aerosol cloud interaction (ACI) in Arctic regions (Coopman et al., 2016; Zamora et al., 2016) and its impact on radiation (Earle et al., 2011; Tietze et al., 2011). One common definition used for quantification purposes is:

$$ACI = \frac{1}{3}\frac{\mathrm{d}\log_{10} N_{tot}}{\mathrm{d}\log_{10} N_a} \tag{4}$$

with N$_{tot}$ the number concentration of cloud droplets and $N_a$ the number concentration of aerosols (Feingold et al., 2001; McComiskey et al., 2009). For observations, ACI is obtained using a linear regression of $\log_{10}$ N$_{tot}$ and $\log_{10} N_a$. We prefer defining ACI using N$_{tot}$ instead of r$_{eff}$, because the latter varies stronger vertically and would require to classify the clouds

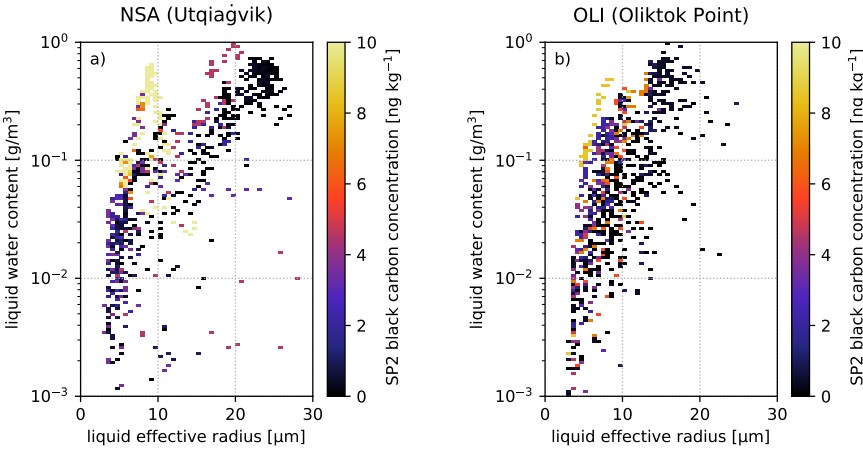

**Figure 8.** As Fig. 4, but with absolute values for SP2 refractory black carbon rBC concentration for (a) OLI and (b) NSA

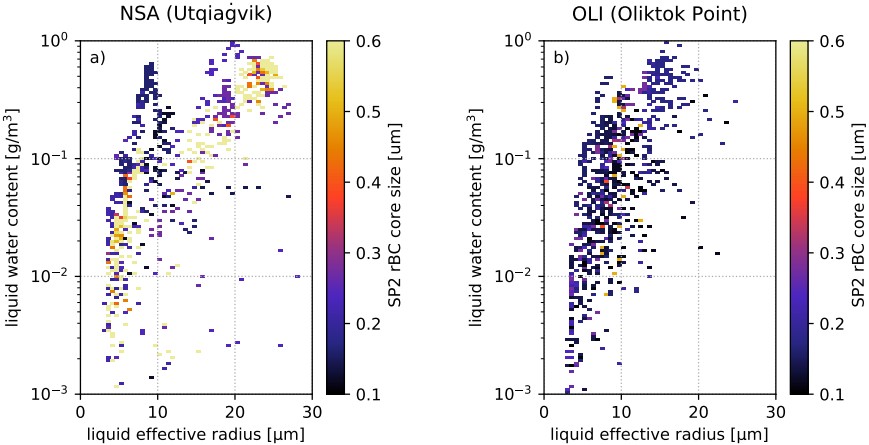

**Figure 9.** As Fig. 8, but with the mean size of refractory black carbon rBC measured below cloud.

by liquid water path, significantly reducing the size of the data set. Fig.14 shows $N_{tot}$ and $N_a$ for both sites. $N_a$ is obtained from the PCASP because it covers the size range of active accumulation mode aerosols best. The ACI value for clouds at both sites is $0.14\pm0.04$ with $R^2 = 0.30$. Even though $R^2$ is small, the ACI value found here is similar to Zamora et al. (2016) who found ACI values of 0.15 for the PCASP using a multi-campaign data set focused on biomass burning. McComiskey and

5    Feingold (2012) found that the choice of platform and observational scales can have a significant impact on the estimation of ACI making comparisons between data sets challenging. Zamora et al. (2016), however, also used cloud-averaged in-situ aircraft observations and as a consequence we expect them to be comparable. When applying the linear regression to the data sets corresponding to the two sites separately, the obtained ACI values differ (Table 1), with OLI having a lower ACI value $(0.12\pm0.05)$ than NSA $(0.20\pm0.07)$. Given the small sample size (24 and 16 cases for OLI and NSA, caused by the PCASP

10   data being quality-flagged for some cases) and the overlap of the uncertainty ranges (obtained from the linear regression), it

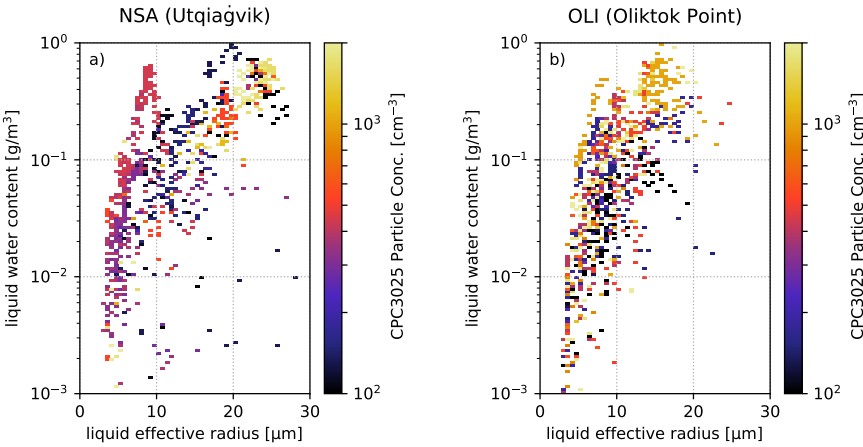

**Figure 10.** As Fig. 8, but with absolute values for CPC3025 condensation nuclei CN concentration.

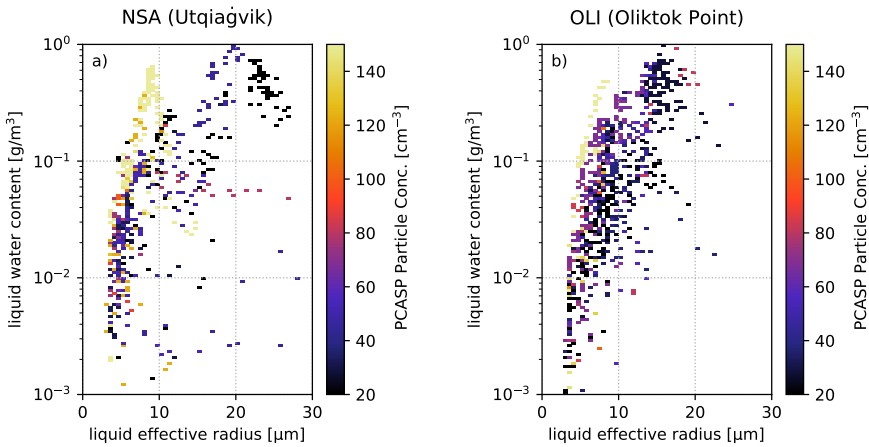

**Figure 11.** As Fig. 8, but with absolute values for PCASP particle concentration.

is not possible to determine whether there is a difference in aerosols nucleation efficiency between the two sites. In addition, given the small sample size, we did not estimate ACI for local clouds only. The lower $R^2$ value for OLI (0.24) in comparison to NSA (0.40) could indicate that the assumption that PCASP particle concentrations represent a good approximation for CCN concentrations is partly violated at OLI. This is consistent with those particles being less aged and consequently less

5  coated by sulphates and organics in comparison to those observed around NSA, though detailed observations of chemical composition were not available for this campaign. In addition, some data points lie above the 1:1 line in Fig. 14 which might indicate that particles smaller than the PCASP size range (i.e. < 100 nm) are acting as CCN (Leaitch et al., 2016). Further, the assumption that the below-cloud aerosol properties govern the cloud microphysical properties might not be true for all clouds depending on sub-cloud vertical mixing. Therefore, we identified all clouds where the above-cloud PCASP concentration is

10  larger than below-cloud (red dots in Fig 14), and indeed half of these clouds are above the 1:1 line. When using the above-cloud

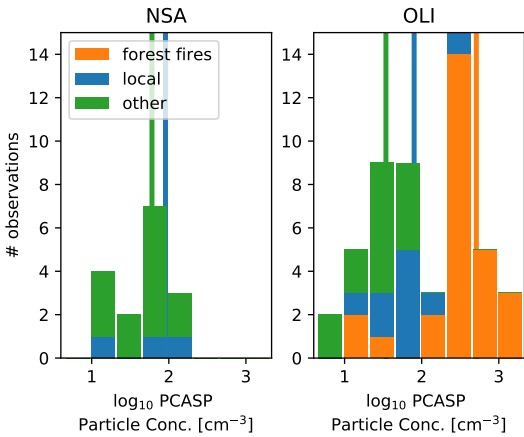

**Figure 12.** Stacked histograms of PCASP particle concentration for a) NSA and b) OLI for clouds classified as forest fire (orange), local (blue) and the residual (green). The vertical lines are for the corresponding linear mean values.

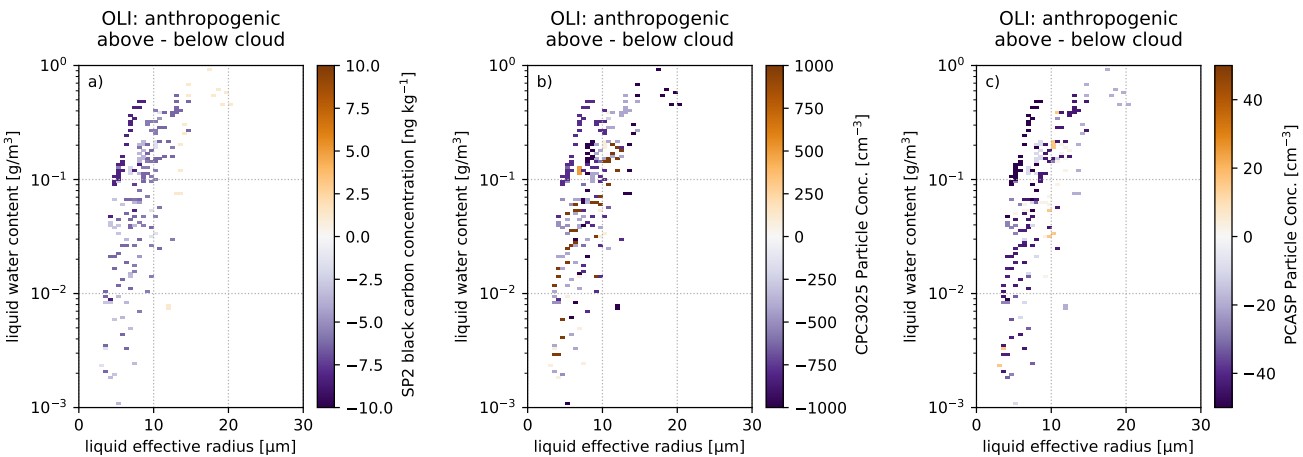

**Figure 13.** As Figs. 8, 10, and 11 for OLI, but showing the difference of rBC (a), CN (b), and PCASP concentration (C) between observations above and below the cloud.

concentration for these clouds, only two of these clouds are above the 1:1 line. However, there are still 11 more clouds above the 1:1 line. Since these clouds generally feature PCASP concentrations $< 50$ cm$^{-3}$, the fact that they are above the 1:1 line could be related to increasing sampling errors for small concentrations, but might also be related to activation of aerosols below 100 nm diameter (Leaitch et al., 2016).

5   For comparison, we also evaluate ACI calculated including data points associated with forest fires. Based on the flight patterns executed, all of the cloud measurements associated with forest fire emissions were sampled in the vicinity of OLI as discussed above. For the measurements collected, aerosols associated with forest fires generally feature higher PCASP concentrations (and in consequence smaller r$_{eff}$), which is consistent with ageing of these particles during transport, and in contrast

**Table 1.** ACI values for the subsets presented in Fig. 14

| Data set | ACI | $R^2$ | # clouds |
|---|---|---|---|
| both sites | 0.14±0.04 | 0.30 | 40 |
| OLI | 0.12±0.05 | 0.24 | 24 |
| NSA | 0.20±0.07 | 0.40 | 16 |
| both sites (with fires) | 0.14±0.02 | 0.47 | 67 |
| OLI (with fires) | 0.14±0.02 | 0.48 | 51 |

to the freshly emitted particles generally found around OLI. As already discussed, clouds associated with local emissions have lower PCASP (and likely accumulation mode) concentrations than forest fires, but still have larger concentrations than for the other clouds. When including cases associated with forest fire emissions, ACI is found to be 0.14±0.02 for both OLI and the complete data set, and is similar to results obtained when omitting forest fire influenced cases. Therefore, we conclude that a difference in ACI between local emissions and forest fires cannot be found, given the limited data set. This refers only to the mechanisms through which aerosols change cloud properties, and does not imply that local emissions do not change cloud properties.

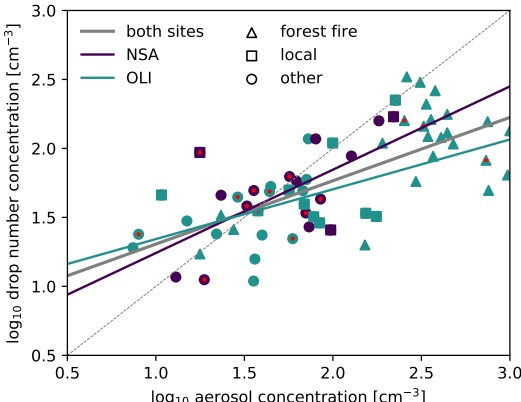

**Figure 14.** Aerosol indirect effect defined using cloud averaged cloud drop concentration $N_{tot}$ and PCASP aerosol concentration $N_a$ obtained below cloud. The colour differentiates between OLI (green) and NSA (purple). Clouds classified as anthropogenically impacted, related to forest fires, and remaining other cloud are marked with square, triangle, and circle, respectively. Red dots indicate clouds where the aerosol concentration above the cloud is higher than below the cloud. The trend lines indicate the linear regressions to obtain ACI (excluding forest fires) for the complete data set (grey), NSA (purple) and OLI (green), the doted line is the 1:1 line.

## 7 Conclusions

The impact of local emissions from industrial activities in northern Alaska on liquid clouds has been investigated based on vertical profiles of aerosol and cloud properties during the ACME-V aircraft campaign, together with measurements from the ARM sites in Northern Alaska: Oliktok Point (OLI) and Utqiaġvik (formerly known as Barrow or ARM's North Slope of Alaska site, NSA). Our main findings can be summarised as follows:

1. Concentrations of condensation nuclei (CN) and refractory black carbon (rBC) are higher in the OLI area (Fig. 3). This is related to emissions associated with local oil and natural gas extraction activities. In contrast, PCASP particle concentrations (diameter > 100 nm, mostly accumulation mode) are not elevated around OLI when compared to NSA.

2. In addition, we found (Fig. 4) that liquid clouds generally feature significantly smaller $r_{eff}$ at OLI when compared with NSA for LWC > 0.1 g kg$^{-1}$. Clouds with $r_{eff}$ > 18 $\mu$m are only rarely observed at OLI. Furthermore, collision-coalescence and precipitation rates are reduced by up to two orders of magnitude around OLI (Fig. 6). Only half of this reduction can be explained by the reduced $r_{eff}$. As a consequence, the breadth of the size distribution of liquid droplets is smaller at OLI as was observed (Fig. 5.b). The reduction of $r_{eff}$ at OLI occurs despite the larger background of PCASP concentrations at NSA (Fig. 3.c)

3. Multiple lines of evidence connect these changes in cloud properties to the observed local emissions. First, HYSPLIT simulations show that 50% of all cloud observations around OLI can be traced back to local emission sources (Fig. 7). Second, some clouds with mid-size $r_{eff}$ (between 9 and 12 $\mu$m) at OLI correspond to increased CN and rBC concentrations (Figs. 8, 10). Third, the mean size of cloud-associated rBC particles is smaller at OLI which is consistent with the assumption of anthropogenic sources (Fig. 9). Finally, the clouds identified as most likely influenced by anthropogenic activities have significantly higher PCASP concentrations for OLI than for the remaining clouds (Fig. 12). However, the PCASP concentration of local clouds is not significantly higher than at NSA which might be related to a higher background of PCASP particle concentrations at NSA.

4. Given the limited data set, we found ten of 24 clouds at OLI, but only one of 16 clouds at NSA which might be impacted by local anthropogenic emissions.

5. The PCASP concentration of clouds associated with forest fires is on average six times larger than for locally impacted clouds (Fig. 12). Consequently, the impact of local emissions on cloud properties is small compared to the influence of forest fires (Fig. 14).

6. Quantification of aerosol cloud interaction (ACI) is challenging due to the small data set. Having said this, based on evaluation of clouds impacted by both local emissions and forest fires, the results are consistent with previous studies of ACI in the Arctic environment (Fig. 14). While forest fire cases have typically higher PCASP concentrations and consequently droplet concentrations, their inclusion into the estimation of ACI does not substantially alter the found relationship.

Because liquid clouds were observed most often (60%), the impact of local pollution on mixed phase and pure ice clouds is not covered here. Moreover, the question as to what percentage of clouds at OLI (and NSA) is impacted by local emissions and whether the industrial activities at the North Slope of Alaska also lead to a change in local climate (e.g. due to cloud radiative forcing, precipitation impacts, or cloud life cycle), cannot be answered with in-situ aircraft measurements alone. These questions can likely better be answered using ground- and satellite-based remote sensing data from OLI and NSA by identifying differences between the sites in cloud cover, liquid water path, emissivity, effective droplet size, and precipitation occurrence. Nevertheless, based on this limited in-situ data set we can conclude that local emissions form industrial facilities in Alaska do influence local cloud properties while the overall spatial extent of these influences has yet to be evaluated. Given the observed cloud modifications, the effects of anthropogenic pollution on local climate should be considered when developing industrial infrastructure in an already fragile and warming Arctic environment.

## 8 Data availability

Data were obtained from the Atmospheric Radiation Measurement (ARM) climate research facility, a U.S. Department of Energy Office of Science user facility sponsored by the Office of Biological and Environmental Research. The surface observations from OLI and NSA as well as the ACME-V data set are available at the ARM archive www.arm.gov/data (see ARM, 1993, updated daily, 2016), the phase classification of the cloud probes is available from the corresponding author on request.

*Acknowledgements.* The authors would like to thank the teams of the ARM facilities (aerial, Oliktok Point, Utqiaġvik/Barrow) and the team responsible for processing and providing the data of the ACME-V campaign: among others S. Biraud, D. Chand, C. Flynn, J. Hubbe, C. Long, A. Matthews, M. Pekour, B. Schmid, A. Sedlacek, S. Springston, J. Tomlinson. Contributions from MM, GD and JC were supported by the US Department of Energy Atmospheric System Research (DOE ASR) program under award number DE-SC0013306. GD was additionally supported by NSF project ARC 1203902. GF was supported under DOE ASR grant No. DE-SC0016275; GM and WW were supported under DOE ASR grant No. DE-SC0016476 (through UCAR subcontract Z17-90029). The authors gratefully acknowledge the NOAA Air Resources Laboratory (ARL) for the provision of the HYSPLIT transport and dispersion model used in this publication. We thank the editor and the two anonymous reviewers for their helpful comments. We also acknowledge valuable discussions with Franziska Glassmeier.

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
