# Peer review of "The observed influence of local anthropogenic pollution on northern Alaskan cloud properties"

_Atmospheric Chemistry and Physics, 2017_

## Referee Comment (RC1) · Anonymous Referee #2 · 19 May 2017

**Review of:**
The observed influence of local anthropogenic pollution on northern Alaskan cloud properties
Authors: Maximilian Maahn et al.
MS No.: acp-2017-302
MS Type: Research article

**General comments:**

This paper is well written, and the figures are mostly easy to understand. I like how they provided the ACI value for their observations, which will be useful for comparison to other locations and aerosol types, and possibly for model parameterization. As best I can tell, the findings are not majorly novel, but it is clear that the authors worked hard to make their findings useful to the community, and there is a lot of good information that would be nice to have available in a publication.

However, I do have a few concerns that need to be addressed before I would recommend this paper for publication. There was repeated mention of various differences between the sites, but I was not satisfied with the lack of discussion of the meaningfulness of these differences (see specific comments below). For the reader to understand and evaluate the author's conclusions, more clarifications, appropriate statistical analysis, and/or assessment of errors is needed. I also disagree with the author's interpretation of the influence of oil-field particles on local clouds. However, perhaps I am just misunderstanding something that will become clearer once additional analysis and/or clarification has been provided.

**Specific comments:**

Most important comments:

1) There were many instances in the text where a better discussion on meaningfulness is needed to support the author's statements. Here are some examples:

> P5l25: "*We manually inspected the vertical profiles of rBC and carbon monoxide (CO), which together constitute a good tracer for biomass burning*" rBC and CO together don't necessarily make good tracers for smoke… anthropogenic combustion processes also create those tracers, and photochemistry affects CO concentrations in the Arctic. Can you discuss the uncertainties related to this statement, and its potential impacts on your results?
> P6l17-19: How meaningful is the difference in skewness at the two sites? Can you demonstrate a statistically significant difference between the skewness, and/or discuss the impact of outliers/sample number/errors, etc.?
> P6l23-24: If you keep this sentence, please provide more details on how meaningful this difference is (including the amount of difference, if it is statistically significant, and if these numbers are trustworthy given that a)

particles of this small size are on the low end of the detectability range for both instruments, and b) that there is typically some error involved in the measurements themselves, and both samplers would ideally be calibrated correctly for these comparisons.)

P6l32-P7l2: "*In contrast to traces from forest fires, carbon monoxide (CO) concentrations were not found to be significantly enhanced in the OLI region (not shown).*" I think some clarification here is needed. Please specify what you mean by traces (tracers?) of forest fires. Do you just mean rBC or are you including CN from the CPC instrument (or MODIS/HYSPLIT information?). Some discussion of the uncertainties in this statement would also be useful. For example, BB smoke can produce large amounts of brown carbon not necessarily detected by a nephelometer; there are many other sources of CN besides forest fires; what CO changes would be constituted as meaningful; were background CO values already high to begin with (which might mask small local changes)?; etc. By "significantly enhanced" do you mean you did some statistical analysis? What was it? In general, I don't think you have presented enough evidence for this statement to be meaningful or useful to the reader yet.

P7l3-7: How do you know this change is significant? I might have missed it, but I didn't see any information presented on sample number.

P7l28: What do you mean by "significant"? If there was a significance test you used to make this statement, what was it?

P8l2-3: Some kind of statistical analysis would be helpful here… at minimum are the differences in means significant? Also, a "shift" implies movement from one state to another, and you haven't presented the information to show that movement occurred. Instead, what you show is that there is a secondary peak at NSA that is not present at OLI. My opinion, based on the information provided, is that one can only speculate on the reasons for this difference and not assign a cause/mechanism.

P10l3-6: "*It can be seen that C is decreased at OLI in comparison to NSA by up to one order of magnitude for constant LWC and $r_{eff}$.*" Since sample numbers are kind of hard to extract from Fig. 5, it might be helpful to mention the sample numbers provided for this comparison. Also, what are the errors in C to begin with? Perhaps it would be more appropriate to compare the overall site values of C, instead of single bin values to reduce random errors and biases from samples from a single cloud dominating one of the bins. At minimum, the authors should qualify this statement with the uncertainties involved. One suggestion is to move Fig 6 to the supplement, and just present mean differences between the sites in the main text. For the following sentence, "*Interestingly, differences in C are largest for $r_{eff}$ smaller than 16 μm*" again, what is the statistical basis for this statement? To me, it looks like all the samples > 16 um have a very small sample number for comparison.

P10l11-12: "$\overline{C}$ *is reduced by 1 to 1.5 orders of magnitude at OLI. The offset is surprisingly constant for LWC larger than 0.01 g $m^{-3}$*" Can you trust the

data at LWC < 0.01 g m-3?  Other studies define LWCs below that value
as not even being in-cloud (e.g., Matsui et al. (2011)).  Do you have
statistical basis for this statement?

P12l21-22: "*since no enhanced PCASP particle concentrations are found to be
correlated to droplet sizes in the emissions-impacted 9 to 12 μm range
(unlike for rBC and CN), there is no indication that local emissions are
directly altering liquid clouds to have smaller $r_{eff}$ as a result of PCASP-
sized particles.*"  Since no correlation was actually shown, the authors
should either show the correlation or its relevant statistics, and discuss its
meaningfulness, or they should use a different word and present some
discussion on the statistical basis for the new statement.

For general discussion of Figs. 8-11: What is the influence of random error? Was
there bias, e.g., from a single outlier cloud containing all the values in one
or more of the bins, etc.?

P16l7-8: Does the statement that, "*collision-coalescence and precipitation rates
are reduced by up to two orders of magnitude around OLI*" still hold after
careful consideration of the impacts of sample number and potential
biases?

2) I did not agree with the author's 3$^{rd}$ conclusion (on page 17), that local emissions
affect cloud properties. Their arguments were as follows:

a) "*HYSPLIT simulations show that 62% of all cloud observations around
OLI can be traced back to local emission sources.*"

First, I am not clear on what was meant by "*cloud observations around OLI can
be traced back to local emission sources*" (see specific comment #17 below).  I do
agree that OLI is more likely to contain oil-field related particles than NSA, based
on proximity and the rBC, CN, and HYSPLIT analyses the authors presented.
However, this argument alone is not enough to indicate that oil-field related
emissions are impacting clouds. The authors present evidence to show that higher
CN at OLI is due to elevated small (<100 nm) particle concentrations, and they
mention that these particles are likely too little to act as CCN. As the authors
mention, rBC is not necessarily a good indicator of CCN either.  Moreover, the
range of below-cloud PCASP aerosol concentrations, which are in the size ranges
where droplet nucleation would be more likely, were similar between sites (Fig.
4).

b) "*Reduced $r_{eff}$ (between 9 and 12 μm) of OLI clouds correspond to
increased CN and rBC concentrations*"

Convincing supporting evidence for the validity and meaningfulness of this trend
has not yet been presented (see specific comment #1).

c) *"the mean size of cloud-associated rBC particles is smaller at OLI which is consistent with freshly emitted, less aged particles"*

There were smaller particles at OLI, and these small particles probably were less aged, but don't the CN and PCASP data indicate that these particles are likely too small to act as efficient CCN?

d) *"clouds [at OLI] were found to be frequently connected to enhanced CN and accumulation mode concentrations"*

I think the authors meant "associated with" instead of "connected to"? But again, the presence of smaller particles at OLI does not necessarily indicate that they are participating in cloud processes. As I mentioned, the range of below-cloud PCASP aerosol concentrations, which are in the size ranges where droplet nucleation would be more likely, were similar between sites (Fig. 4). As the authors mentioned in the text, particles <100 nm in diameter are likely to be less efficient CCN.

In summary, there wasn't convincing evidence that oil-field aerosols did have a discernable impact on cloud properties.

More minor comments (not in order of importance):

3) P2l23: It might be helpful to note upfront that liquid clouds are not that common in the Arctic, compared to ice and mixed phase clouds?
4) P5l23: "*While Arctic Haze was not observed during ACME-V, transported emissions from forest fires can contribute significantly to summertime aerosol loading in the Arctic…*" Earlier Arctic haze was defined as long-range transport of aerosol particles from lower latitudes. Here it seems like the long-range transport of smoke is excluded from consideration as Arctic haze. I think it would help to clarify this sentence, especially since smoke does contribute to what most people define as Arctic haze (e.g., Warneke (2010)). Also, how do you know that what you are referring to here as Arctic haze (Anthropogenic pollution from lower latitudes? Long-range aerosol transport?) was not observed during ACME-V? Is this statement based on the fact that typical precipitation and air mass transports during the summer season make the presence of long-range aerosol transport less likely, or on some extra analysis you did to determine this? Please specify.
5) Last paragraph of section 2: This paragraph is a little confusing, and it would help if the authors were more clear about their methods. What was the order and/or priority of the steps to ID long-range smoke transport? From where were the CO data obtained? It says the vertical profiles of rBC and CO were manually inspected – but what were the criteria relating to those parameters, and what were those criteria specifically used to do? Were the 5 locations where HYSPLIT back trajectories were obtained chosen based on MODIS fire locations?

6) Fig1: Maybe include the MODIS fire locations?

7) P6l11-12: "*The data presented are limited to observations obtained below 500 m in order to demonstrate the impact of local emissions and reduce the impact of forest fires.*" Up to this point in the text, the reader has not been presented with any strong evidence that smoke (or long-range anthropogenic pollution, for that matter) was not present below 500m. To avoid confusing the reader, please provide this information.

8) P6l29-31: It would be helpful to use consistent units here to aid comparisons with the data in this study. Also, the rBC concentrations referenced from Zamora et al., were for background air masses specifically thought to not be influenced by smoke (see their Table 6). From their Figure 7, smoky rBC values ranged somewhere between the order of $1\text{-}10^3$ ng/m$^3$ (average appears on the order of $10^2$ ng/m3). Also, the majority of rBC values presented in Figure 3 are much below the median 20-30 ng/kg values from Roiger et al. So what I interpret the rBC from Figure 3 to mean is that the region was very clean with respect to rBC, except near OLI.

9) P7l7-13: Your expectation that that there might be more smoke influence at the NSA site would be consistent with literature observations of general smoke particle sizes being larger than the PCASP minimum size range of ~100 nm (e.g., Kondo et al. (2011); Sakamoto et al. (2015)). However, I am confused by your statement that, "*An alternative explanation could be … more aerosol processing by precipitation (e.g. Hoppel et al., 1990).*" I believe that the Hoppel et al. reference refers to non-precipitating cloud processing of aerosols, correct? So do you mean "processing by clouds" instead of "processing by precipitation"? In which case, why did you mention collision-coalescence and precipitation being the causes for the differences? Or did you mean "removal" or "scavenging" here instead of "processing" (in which case why the subsequent sentence regarding aerosol in-cloud processing)? Other readers might be confused as well, so clarification here would be helpful.

10) P7l18: "*While mean particle size generally increases with decreasing CN concentration, …*" Looking that figure 4, this is not clear to me. I suggest taking this statement out, since it doesn't seem that necessary. Or, if you disagree, you could try plotting on a log-scale to see if the pattern emerges more clearly or switching the x and z axes to show the trend better?

11) P7l19-20: "*the variability of PCASP mean size is rather low for CN concentrations > 600 cm$^{-3}$, which is consistent with the idea that particles have already experienced growth*" Are there other things that might cause the same trend? Might be worth mentioning.

12) Fig.4: why was the y-axis cut off at 4000? To provide aerosol context for the cloud data? To provide better context for the study as a whole, you might consider plotting on a log scale to include all the data, or at least mentioning in the caption that a fraction of the data are excluded for whatever your reason.

13) P9l6-7: "*It is not possible to study the cloud life cycle using aircraft in-situ observations, but the potential for impact on cloud life cycle can be estimated by…*" I suggest moving your qualifiers in the last paragraph of this section up here

so that you can better clarify upfront that what you are estimating is a factor important to cloud lifetime, but your estimate is not necessarily indicative of cloud lifetime itself.

14) P10l4-5: "*This is caused by reduced broadening of the drop size distribution towards large drops at OLI (not shown), consistent with cloud chamber experiments (Gunn and Phillips, 1957).*" Maybe I am not understanding something, but shouldn't that kind of broadening be apparent in Fig. 5? Why not mention that here? Also, what specifically is consistent with this cloud chamber experiment?

15) P10l9: "*typical $r_{eff}$ values are reduced at OLI in comparison to NSA for the same LWC.*" I don't see that consistently in Fig. 5. What about at high LWCs, for example? Did you mean within a specific LWC range? If so, please specify.

16) P10l13: more info on this droplet fall velocity parameterization would be helpful.

17) P11l6: "*For OLI (NSA), 62% (16%) of all ACME-V cloud observations can be traced back to surface emissions originating from the Prudhoe Bay oilfields.*" I am confused here. How did were these values calculated? What does it mean that a certain fraction of cloud observations were traced back to surface emissions? That some fraction of the aerosol particles in the air masses containing the clouds were likely to have originated from the oilfields? Please clarify in the text.

18) Fig. 7 and text citing it: It is not clear to me how one can get the mass concentration at a certain location unless the emissions at the point source are known. Is this a relative mass concentration? Was emissions information available? If so, please provide it. Also, what are the errors in the co-location of the modeled plume and the actual plume? Please discuss. If you don't have this information, I recommend removing this figure and just stating in the text that HYSPLIT back trajectories indicate a higher influence of oil field aerosols at OLI than at NSA x% of the time, and that one would expect higher oil field aerosol concentrations at OLI than at NSA due to proximity of the source. Figs. 8 and 9 probably provide more trustworthy information on aerosol concentrations anyways (please correct me if I am wrong).

19) P16l8-9: Regarding the statement that, "*As a consequence, the breadth of the size distribution of liquid droplets has to be smaller at OLI*", why not just provide the size distribution data from the campaign?

20) p. 17l10-12: "*While forest fire cases have typically higher aerosol concentrations and consequently droplet concentrations, their inclusion into the estimation of ACI does not substantially alter the found relationship.*" I disagree with this statement. Based on the arguments from specific comment #2 above, the authors admission that they might not have been able to completely exclude smoke-influenced air masses from the study, and the similar ACI values for smoke influenced cases, for all we know, smoke was driving the relationships… or did I miss something?

**Technical corrections:**

P3l15: You cite a 1993 paper for 2015 observations. Please clarify what you meant by citing this paper here.

P6l15: Just a suggestion, get rid of "For the SP2", since people may not remember what SP2 is, and it is not necessary to the sentence.

P6l20: Suggest combining this sentence with the following sentence to make it clear what specific pattern(s) is/are being deemed similar (the skewness and spatial distribution as opposed to the median?).

P6l2-4: The units in this sentence are confusing. Also, can you clarify in the manuscript that the latter two numbers are for wet deposition scavenging (or some other kind of scavenging)?

P8l1: suggest rewrite to "*As mentioned above, clouds **known to be** impacted by forest fires have been removed*" to better reflect the uncertainty you discussed in the previous section.

Fig. 5: please describe in the caption what the dashed line indicates. Also suggest adding in the term "NSA", since that is what you talk about in the text referring to this figure. Also, I found the label "number of observations [%]" to be very confusing. I recommend changing this to something like, "% of total observations."

Fig.6: The caption says OLI-NSA, but the figure says NSA-OLI? Also, can you make the axes consistent between Fig. 5 and 6? That would help enable comparisons between the two. Lastly, in the caption it says that, "*The green dots highlight data points with less than five observations.*" Please clarify in the caption whether those 5 observations include points at both sites added together, or at each site individually.

Suggest using site abbreviations throughout the paper and in the figures after they have been defined, instead of using a mix of the site names and the abbreviations to avoid confusion.

P12l5: Please clarify: what "this" do you refer to when you say, "This means…"?

Fig 12: concentration not cooncentration on the x-axis.

P17l19: from not form.

**References**

Kondo, Y., Matsui, H., Moteki, N., Sahu, L., Takegawa, N., Kajino, M., Zhao, Y., Cubison, M. J., Jimenez, J. L., Vay, S., Diskin, G. S., Anderson, B., Wisthaler, A., Mikoviny, T., Fuelberg, H. E., Blake, D. R., Huey, G., Weinheimer, A. J., Knapp, D. J. and Brune, W. H.: Emissions of black carbon, organic, and inorganic aerosols from biomass burning in North America and Asia in 2008, J. Geophys. Res. Atmospheres, 116(D8), D08204, doi:10.1029/2010JD015152, 2011.

Matsui, H., Kondo, Y., Moteki, N., Takegawa, N., Sahu, L. K., Zhao, Y., Fuelberg, H. E., Sessions, W. R., Diskin, G., Blake, D. R., Wisthaler, A. and Koike, M.: Seasonal variation of the transport of black carbon aerosol from the Asian continent to the Arctic

during the ARCTAS aircraft campaign, J. Geophys. Res. Atmospheres, 116(D5), doi:10.1029/2010JD015067, 2011.

Sakamoto, K. M., Allan, J. D., Coe, H., Taylor, J. W., Duck, T. J. and Pierce, J. R.: Aged boreal biomass-burning aerosol size distributions from BORTAS 2011, Atmos Chem Phys, 15(4), 1633–1646, doi:10.5194/acp-15-1633-2015, 2015.

Warneke, C., Froyd, K. D., Brioude, J., Bahreini, R., Brock, C. A., Cozic, J., de Gouw, J. A., Fahey, D. W., Ferrare, R., Holloway, J. S., Middlebrook, A. M., Miller, L., Montzka, S., Schwarz, J. P., Sodemann, H., Spackman, J. R. and Stohl, A.: An important contribution to springtime Arctic aerosol from biomass burning in Russia, Geophys. Res. Lett., 37, 2010.

---

## Referee Comment (RC2) · Anonymous Referee #3 · 27 Jun 2017

The authors use airborne observations from June-September, 2015 within 90 km of two DOE-ARM sites (North Slope Alaska or NSA and Olitok Point or OLI) to study potential effects of changes in aerosol particles on Arctic liquid water clouds. The aerosol observations are limited to physical measurements with a PCASP (particles larger than about 100 nm) and a condensation particle counter (CPC; particles large than about 3 nm as well as measurements of black carbon (rBC). The main objective is to see if clouds formed on particles from anthropogenic activities in the OLI area result in differences in cloud microphysics compared with cloud formed on particles observed in the NSA area. The topic is relevant for ACP, the paper is well organized and interesting, and I believe the overall results are useful. However, the paper is not

currently ready for publication.

Major comments:

1) The paper leads up to section 6 (Quantification of cloud aerosol interaction) by accumulating information suggesting the microphysics of the OLI clouds are impacted by local emissions. That notion is then set aside in section 6 based on the calculated ACI indices. However, the ACI calculation in inappropriate for these observations if particles smaller than 100 nm are nucleating droplets. Drawing a 1:1 line in Figure 12 indicates some points above. Assuming the measurements are reasonably accurate, which the authors do not discuss, then relatively few droplets are nucleated on particles smaller than the lower limit of the PCASP. The larger deviations above the 1:1 line are towards lower aerosol concentrations, which would be consistent with Leaitch et al. (ACP, 2016) if there are sufficient particles smaller than 100 nm to explain the deviations. As it stands, the ACI discussion tells us only that there is some impact of PCASP-sized particles on the NSA cloud observations, which has already been mentioned in connection with Figure 10 and is not the focus of the paper. The fundamental result could be more clearly shown using Figure 12 with straight concentrations rather than natural logarithms. What do Figures 8, 9, 10 and 11 tell us that cannot be found from a modified Figure 12?

2) Concerning Figure 6 and related discussion on page 10, I have the following questions and remarks:

a) The differential collection growth rates (OLI minus NSA) range between 0.1 and 1 (units of kg/sm3). The mean collection growth rates vary from less than $10^{-12}$ to about $5 \times 10^{-6}$ with the same units. How can the differential rates be higher than the mean rates? Are the differential values ratios (e.g. OLI-NSA/NSA) rather than absolute values?

b) On lines 3-4, you say that "C" is lower at OLI compared with NSA for constant LWC and Reff. This is very difficult to see in 6a. Regardless of whether the differentials

(OLI-NSA) are absolute or ratios, they are higher, not lower. Also, how do I look at constant LWC and Reff in these plots?

c) On lines 10 and 11, you say that the mean value of C (averaged for Reff) is 1 to 1.5 orders of magnitude reduced at OLI for a constant LWC. Yet the mean OLI growth curve lies to the right of the NSA curve, which indicates a higher mean collection growth rate.

d) The same apparent discrepancies are present for the rainfall rates in Fig. 6b.

3) The aerosol observations are restricted to below 500 m-msl. Please indicate the altitude range for the cloud observations. Please indicate how you know that the aerosol below cloud was connected with the cloud above and not isolated by temperature inversions, which can happen in the relatively stable environment of the Arctic.

Minor comments:

4) Page 2, lines 4-5 – Aerosol number concentration or mass concentration? If number, what sizes? This statement is very simplistic.

5) Page 3, line 18 – what are "bulk" probes?

6) Pages 3 and 4 – If not described, references are needed for how the CDP, DCDP and OAPs were evaluated and calibrated during the study.

7) Page 4, line 6 – A droplet threshold of 10/cc may not be appropriate for Arctic summer clouds (e.g. Leaitch et al., ACP, 2016), and it does not allow consideration of situations such as discussed by Mauritsen et al. (ACP, 2011). Please discuss.

8) Page 4, line 10 – what are "tiny" particles?

9) Page 5 – Describe the inlet for particles measured with the CPCs.

10) Page 5, line 11 - How was the PCASP calibrated during the study? Is the lower detection limit truly 100 nm (e.g. Liu et al.: Response of Particle Measuring Systems airborne ASASP and PCASP to NaCl and latex particles, Aerosol Sci. Technol., 16,

83-95, 1992)?

11) Page 5, line 20 - How as the SP2 calibrated?

12) Page 5, line 23 – While Arctic Haze is not common during the summer, how can you be certain it was not observed? PCASP number concentrations of 150-200/cc may be representative of Arctic Haze (e.g. Leaitch et al., J. Atmos. Chem, 9, 187-211, 1989).

13) Page 5, line 35 – A particle density of 6 g/cm3 is very high. The density of submicron particles is usually less than 2.5 g/cm3. Please explain.

14) Page 6, line 12 – aerosol data

15) Page 6, line 23 – reference for size of freshly emitted soot?

16) Page 6, line 26 - qualify "quickly" by assuming sufficient gaseous precusors.

17) Page 7, line 22 – smallER

18) Page 7 - In Fig. 4, there are interesting similarities between the small group of yellow points associated with each site. Both groups show increases in PCASP size with increasing CPC. This curious group is related to cloud for the OLI case but not the NSA case. Can you identify a connection?

19) Page 9, line 1 - Indicate the reason for the 16 um line in the caption of Fig. 5. Also, in Fig. 5, please add a line showing how LWC vs LER varies assuming the mean droplet number concentration for each location. What are those mean droplet concentrations?

20) Page 12, lines 1-3 – Is there evidence that OLI emissions impacted any of the NSA observations?

21) Page 12, lines 9-10 - Reduced Reff with increased rBC is not so clear; these plots have a qualitative aspect to them. At the higher LWC (>0.1 g/m3) that may be true, but

below 0.1 it appears that the opposite may be true.

22) Page 12, lines 15-16 - When averaged over a large number of observations. Also, the "notion" is commonly anticipated for clouds with higher LWC (roughly >0.1 g/m3; e.g. Leaitch et al., JGR, 1992) when effects of evaporation, dissipation and precipitation are reduced factors.

23) Page 12, Lines 18-20 – The use of monotonic is not justified here.

24) Page 13, lines 8-9 - How many CCN are needed for cloud formation?

25) Page 15, header for section 6 – You are not discussing an "interaction" here, only a potential impact of the aerosol on the cloud.

---

## Author Comment (AC1) · 23 Aug 2017

Please see the attached supplement.

Please also note the supplement to this comment:
https://www.atmos-chem-phys-discuss.net/acp-2017-302/acp-2017-302-AC1-supplement.pdf

———————————————

---

## Author Response (AR1)

**The observed influence of local anthropogenic pollution on northern Alaskan cloud properties**

**Response to reviewers**

M. Maahn, G. de Boer, J. Creamean, G. Feingold, G. McFarquhar, W. Wu, and F. Mei

August 23, 2017

Original Referee comments are in italic

manuscript text is indented, with added text underlined and <del>removed text</del> erossed out.

We would like to thank the reviewers for their detailed and helpful comments. We revised the manuscript and responded to all of the reviewers' comments. In addition to the reviewers' suggestions, we modified:

- We removed Fig. 4 in order to shorten the manuscript and because it did not contribute to the main focus of the article.
- The OLI data set included some clouds sampled directly after aircraft take-off. Due to their low cloud base (as low as 23 m) they were more likely to be impacted by surface emissions and biased the comparison to NSA where the lowest cloud base sampled was 178 m. Therefore we removed all clouds less than 3 km in distance to the airport which increased the minimal cloud base for OLI to 221 m. We added:

Data obtained during take off and landing have been removed to avoid skewing the comparison by sampling aerosols and clouds at much lower altitudes than elsewhere.

- We refer to aerosol observations of the PCASP no consistently to as 'PCASP concentration' and avoid the use of the term 'accumulation mode concentration', because the PCASP does not cover the full accumulation mode size range.
- Several smaller changes, mostly with respect to grammar and wording. Please see the attached pdf of the manuscript with all text changes highlighted for details.

Please note that a companion paper to this study has just been published in ACP Discussions (Creamean et al., 2017).

**1 Reviewer III**

**General comments:**

This paper is well written, and the figures are mostly easy to understand. I like how they provided the ACI value for their observations, which will be useful for comparison to other locations and aerosol types, and possibly for model parameterization. As best I can tell, the findings are not majorly novel, but it is clear that the authors worked hard to make their findings useful to the community, and there is a lot of good information that would be nice to have available in a publication.

However, I do have a few concerns that need to be addressed before I would recommend this paper for publication. There was repeated mention of various differences between the sites, but I was not satisfied with the lack of discussion of the meaningfulness of these differences (see specific comments below). For the reader to understand and evaluate the author's conclusions, more clarifications, appropriate statistical analysis, and/or assessment of errors is needed. I also disagree with the author's interpretation of the influence of oil-field particles on local clouds. However, perhaps I am just misunderstanding something that will become clearer once additional analysis and/or clarification has been provided.

Specific comments:

Most important comments:

1) There were many instances in the text where a better discussion on meaningfulness is needed to support the author's statements. Here are some examples:

P5l25: "We manually inspected the vertical profiles of rBC and carbon monoxide (CO), which together constitute a good tracer for biomass burning" rBC and CO together don't necessarily make good tracers for smoke? anthropogenic combustion processes also create those tracers, and photochemistry affects CO concentrations in the Arctic. Can you discuss the uncertainties related to this statement, and its potential impacts on your

**results?**

We agree that the description was too brief. We extended:

Therefore, we manually inspected the vertical profiles of rBC and earbon monoxide (CO) CO, which together constitute a good tracer for biomass burning (Warneke et al., 2009, 2010). Typically, these layers are found aloft (Roiger et al., 2015), allowing us to use vertical profiles obtained by the aircraft to aid in their identification. For each spiral obtained at the two sites, elevated layers with CO  $\geq 0.1$  ppmv or rBC  $\geq 20$  ng kg-1 were flagged as corresponding to forest fires. Local emissions, on the other hand, are expected to be found in a layer connected to the surface.

We are confident that our approach to remove forest fires from the data set is feasible: As shown in Creamean et al. (2017) and stated in the discussion of Fig 3, we did not find enhanced CO concentrations in the OLI region. Further, rBC values were increased in the OLI region, but the increase was small in comparison to forest fires. Finally, local emissions are expected to be connected to the surface while transported emissions are found aloft. To make this more clear, we added to the discussion of Fig. 3

The differences between CO and rBC concentrations attributed to forest fires and the concentrations measured in the OLI region show that our approach to use CO and rBC to separate observations impacted by forest fires is feasible

To make clear that we might have removed too many clouds, we added to the end of the section:

For the spirals, data identified as originating from forest fire either from manual inspection or according to HYSPLIT, were removed from subsequent analysis. With this approach, we likely removed more clouds from the analysis than required. This, however, ensures that the analysis of the remaining clouds is not biased by influences from forest fires.

P6117-19: How meaningful is the difference in skewness at the two sites? Can you demonstrate a statistically significant difference between the skewness, and/or discuss the impact of outliers/sample number/errors, etc.?

We replaced the violin plots with real histograms so that the reader can see outliers better. In order to test more features than only the skewness, we applied the two sample Kolmogorov–Smirnov test in order to show that the distributions are significantly different. The text has been modified accordingly.

For both instruments, the distributions within the 90 km circle belonging to each site are skewed towards higher concentrations and the distributions of both sites are significantly different (1% confidence interval) according to the two sample Kolmogorov–Smirnov (KS) test (Massey, 1951)

P6l23-24: If you keep this sentence, please provide more details on how meaningful this difference is (including the amount of difference, if it is statistically significant, and if these numbers are trustworthy given that a) particles of this small size are on the low end of the detectability range for both instruments, and b) that there is typically some error involved in the measurements themselves, and both samplers would ideally be calibrated correctly for these comparisons.)

We added this quantity to Figure 3 so that the reader can evaluate the difference by him/herself. We also state that the uncertainty might be enhanced:

Further, the difference between both the two. CPC instruments, which depends on equates to the concentration of CN between 3 and 10 nm diameter, is enhanced east of OLI (not shown). in the OLI region and the distribution is significantly (KS-test) different to the one at NSA (Fig. 3.c). Because this quantity is stemming from the difference in two instruments at the limit of their measurement range, the data is used here only in a qualitative way. Freshly emitted soot has been shown to be larger than this (> 20 nm), so this range is 15 nm (Zhang et al., 2008), so particles in the 3 to 10 nm size range are likely due to in situ nucleation of aerosol particles from gas phase precursors (i.e., formation of new particles as compared to secondary aerosol formation, where gases condense onto preexisting aerosol, Kulmala et al., 2012). Nucleated aerosols typically have sizes below 3 nm, but quickly grow via condensation and coagulation to sizes > 3 nm (Colbeck and Lazaridis, 2014). This source of nucleated aerosol particles from petroleum and gas extraction activities has been reported by Kolesar et al. (2017) for emissions transported from OLI to NSA.

Regarding CPC calibration, we added additional information to the instrument section:

CPC calibration activities included verifying inlet flow rate with a low pressure-drop bubble flow meter, and determining the size-dependent particle counting efficiency, according to methods defined in Hermann et al. (2007) and Mordas et al. (2008).

P6l32-P7l2: "In contrast to traces from forest fires, carbon monoxide (CO) concentrations were not found to be significantly enhanced in the OLI region (not shown)." I think some clarification here is needed. Please specify what you mean by traces (tracers?) of forest fires. Do you just mean rBC or are you including CN from the CPC instrument (or MODIS/HYSPLIT information?). Some discussion of the uncertainties in this statement would also be useful. For example, BB smoke can produce large amounts of brown carbon not necessarily detected by a nephelometer; there are many other sources of CN besides forest fires; what CO changes would be constituted as meaningful; were background CO values already high to begin with (which might mask small local changes)?; etc. By "significantly enhanced" do you mean you did some statistical analysis? What was it? In general, I don't think you have presented enough evidence for this statement to be meaningful or useful to the reader yet.

Regarding CO, we removed the word 'significant'. A detailed analysis of CO is out of the scope of this article, please see Creamean et al. (2017) for more information about CO observatiosn during ACME V and the difference between forest fires and local emissions. Furthermore, we replaced 'traces' with 'air masses'. Regarding black carbon, please note that this quantity was measured with a SP2 which is designed such that it is not impacted by coating of the particles. We modified

In contrast to traces CO concentrations sampled in air masses originating from forest fires, earbon monoxide (CO) concentrations were not found to be significantly enhanced low altitude CO concentrations in the OLI region were not enhanced relative to background values (Creamean et al., 2017). The differences between CO and rBC concentrations attributed to forest fires and the concentrations measured in the OLI region (not shown) show that our approach to use CO and rBC to separate observations impacted by forest fires is feasible.

P7l3-7: How do you know this change is significant? I might have missed it, but I didn't see any information presented on sample number.

We added the number of observations to Figure 3 and also applied a Kolmogorov–Smirnov test to show the significance of the difference of the distributions. We added

... and the difference in the distributions is significant according to the KS-test with 1% confidence interval .

P7l28: What do you mean by "significant"? If there was a significance test you used to make this statement, what was it?

We removed the word 'significant'.

P812-3: Some kind of statistical analysis would be helpful here? at minimum are the differences in means significant? Also, a "shift" implies movement from one state to another, and you haven't presented the information to show that movement occurred. Instead, what you show is that there is a secondary peak at NSA that is not present at OLI. My opinion, based on the information provided, is that one can only speculate on the reasons for this difference and not assign a cause/mechanism.

We reworded the sentence to replace the word "shift":

When comparing 2D histograms of liquid effective radius and liquid water content for OLI and NSA (Fig. 4, a, b), a shift towards OLI values are shown to feature smaller reff – can be clearly seen in the measurements obtained in close proximity to OLI for the same LWC.

In order to show for what range of LWC values the difference of reff is significant, we added the new Figure 5:

Fig. 5.a reveals that this difference is statistically significant for most LWC  $> 0.1 \text{ g m}^{-3}$  according to a Welch's t-test (Welch, 1947) with a 5% confidence interval (Note that a 5% confidence interval is always used in this study unless noted otherwise). [...] For TOI, the mean  $r_{eff}$  is significantly different than those at NSA for LWC  $> 0.02 \text{ g m}^{-3}$ .

P10l3-6: "It can be seen that C is decreased at OLI in comparison to NSA by up to one order of magnitude for constant LWC and reff." Since sample numbers are kind of hard to extract from Fig. 5, it might be helpful to mention the sample numbers provided for this comparison.

Please see the caption of Fig 4 for the total number of observations, we also added that information to the plot title. We think that with the updated colormap regions with e.g. less then 1% data (corresponding to 9-16 samples) can be easily identified.

Also, what are the errors in C to begin with? Perhaps it would be more appropriate to compare the overall site values of C, instead of single bin values to reduce random errors and biases from samples from a single cloud dominating one of the bins. At minimum, the authors should qualify this statement with the uncertainties involved. One suggestion is to move Fig 6 to the supplement, and just present mean differences between the sites in the main text. For the following sentence, "Interestingly, differences in C are largest for reff smaller than 16  $\mu$ m" again, what is the statistical basis for this statement? To me, it looks like all the samples i 16 um have a very small sample number for comparison.

Regarding the uncertainty of C, we added:

Note that our approximation does not consider the impact of turbulence and droplet charge on C. This might lead to considerable uncertainties, which have—to the authors' best knowledge—not been fully quantified. Because we are interested how C is modified in the OLI region, we show the difference focus on the ratio of C between both sites in–determined at NSA and OLI which should reduce the uncertainty of C. Fig. 6 shows the ratio between NSA and OLI of C as a function of  $r_{eff}$  and LWC.

Regarding the Figure, we decided to keep it in the manuscript, because a comparison of overall site values for C would not work given that C spans easily more than 10 magnitudes depending on LWC and reff (see text). Therefore, a general comparison would

be dominated by differences of LWC and reff. However, we agree that the statement regarding C for reff > 16 um is based on poor statistics and removed it.

P10l11-12: "C is reduced by 1 to 1.5 orders of magnitude at OLI. The offset is surprisingly constant for LWC larger than 0.01 g m-3" Can you trust the data at LWC < 0.01 g m-3? Other studies define LWCs below that value as not even being in-cloud (e.g., Matsui et al. (2011)). Do you have statistical basis for this statement?

In accordance with other studies (e.g. Lance et al., 2011; Hobbs and Rangno, 1998), we decided to use the number of drops for cloud definition, mostly because this is the quantity directly observed by the cloud probes. Our threshold of 10 drops per cm3 (which was criticized as too high by the other reviewer) translates for our data set into a minimum LWC of approx 0.001 g/m3. In order to show the full data set, we decided to show LWC values as little as 0.001 g/m3. However, we agree that we have to explain that and added to the discussion of Fig. 4:

Note that LWC values below 0.01 g m-3 are defined as not in-cloud by some studies (e.g., Matsui et al., 2011; Leaitch et al., 2016), but we decided to show the full data set because the in-cloud definition used here (> 10 droplets cm-3) can result in LWC as low as 0.001 g m-3 and we wanted to make sure that all cloud data points are included in the analysis.

P12l21-22: "since no enhanced PCASP particle concentrations are found to be correlated to droplet sizes in the emissions-impacted 9 to 12  $\mu$ m range (unlike for rBC and CN), there is no indication that local emissions are directly altering liquid clouds to have smaller reff as a result of PCASP-sized particles." Since no correlation was actually shown, the authors should either show the correlation or its relevant statistics, and discuss its meaningfulness, or they should use a different word and present some discussion on the statistical basis for the new statement.

We reworded that sentence:

The fact that the response of  $r_{eff}$  to PCASP aerosol concentrations is very similar is for constant LWC—almost monotonic for both sites is likely because the PCASP covers the aerosol size range most relevant to droplet nucleation . It should be noted that the monotonic decrease in PCASP concentration with increasing droplet size and is consistent with the first indirect effect. However, since no enhanced PCASP particle concentrations are found to be correlated to droplet sizes in the emissions-impacted 9 to 12  $\mu$ m range (unlike for rBC and CN), there is no indication that local emissions are directly altering liquid clouds to have smaller  $r_{eff}$  A different behaviour would indicate that clouds react differently to the same PCASP concentration (which covers most of the accumulation mode size range, see also Sec. as a result of PCASP-sized particles. Even 6).

We added a new Figure (12) to the manuscript in order to discuss the impact of PCASP concertation on cloud properties better:

Analysis of the relationship between clouds and HYSPLIT concentrations, rBC and CN shows that there are some, but not all, clouds at OLI impacted by local pollution. Because enhanced rBC and CN concentrations are expected to be good indicators of anthropogenic activity, they are used to isolate clouds impacted by anthropogenic emissions. Clouds, whose mean below-cloud rBC or CN concentration is above the median concentrations shown in Fig. 3 (4.1 ng kg-1 and 1122 cm-3, respectively), are identified as potentially impacted. When using this criterion, 10 of 24 (3 of 16) clouds at OLI (NSA) are identified as potentially locally influenced (Fig. 12). For NSA, two of the three clouds corresponded to either extremely low CN ( $< 20 \text{ cm}^{-3}$ ) or rBC (< $1 \text{ ng kg}^{-1}$ ) values, making a connection to anthropogenic activities unlikely. But, the clouds classified as anthropogenic at OLI correspond mostly to enhanced concentrations of rBC (Fig. 8) and CN (Fig. 10) and mid-sized roff  $(5 - 15 \ \mu m)$ . Fig. 12 shows how the PCASP concentrations of the potential locally affected clouds compares to the clouds classified as affected by forest fires (these clouds are removed in all other Figures except Fig. 14) and the remaining, non-classified clouds referred to as 'other'. Note that forest fire emissions were also present in the vicinity of NSA, but cloud measurements from these time periods did not pass the quality control measures implemented (continuously ascending or descending profiles). It is striking that the clouds classified as associated with forest fire have a significant (t-test), 6-times larger linear mean PCASP concentration than the clouds classified as locally affected at OLI (510 vs  $80 \text{ cm}^{-3}$ ). Despite this big difference, clouds classified as locally affected still feature PCASP concentrations significantly larger than the clouds classified as other  $(35 \text{ cm}^{-3})$ . We conclude that CN and rBC particles, which were used to classify local clouds, have the potential to grow to accumulation mode particles measured by the PCASP. For NSA, however, the mean PCASP concentration for clouds classified as other is not significantly (t-test) different from the clouds classified as locally affected at OLI. This is also true when including the three clouds classified as locally affected (from which only one is potentially local as discussed above). This is consistent with the findings of Fig. 3 which shows that the general PCASP concentration background is enhanced at NSA in comparison to OLI.

For general discussion of Figs. 8-11: What is the influence of random error? Was there bias, e.g., from a single outlier cloud containing all the values in one or more of the bins, etc.?

We agree that Figs. 8-11 are potentially affected by outliers. Consequently, we reduced the size of the bins such that most pixel show only observations of a single data point. By this, the trend and the variability of the data can be seen. We added:

The bin sizes in Fig. 7 were reduced in order to investigate the variability between clouds. This reveals that only a subset of clouds is associated with local pollution according to HYSPLIT.

P1617-8: Does the statement that, "collision-coalescence and precipitation rates are reduced by up to two orders of magnitude around OLI" still hold after careful consideration of the impacts of sample number and potential biases?

We applied a t-test to show for which LWC values the difference is significant:

Doing so reveals that, for constant LWC,  $\bar{C}$  is reduced by 1 to 1.5 orders of magnitude at OLI. The offset is significant and surprisingly constant for LWC larger than 0.01 g m-3. [...] Averaging over all reff enhances the effect and leads to differences of up to two orders of magnitude for R as a function of LWC. This effect is statistically significant for LWC > 0.02 g m-3.

Please see above for a discussion of uncertainties of C.

2) I did not agree with the author's 3rd conclusion (on page 17), that local emissions affect cloud properties. Their arguments were as follows:

a) "HYSPLIT simulations show that 62% of all cloud observations around OLI can be traced back to local emission sources." First, I am not clear on what was meant by "cloud observations around OLI can be traced back to local emission sources" (see specific comment #17 below). I do agree that OLI is more likely to contain oil-field related particles than NSA, based on proximity and the rBC, CN, and HYSPLIT analyses the authors presented. However, this argument alone is not enough to indicate that oil-field related emissions are impacting clouds. The authors present evidence to show that higher CN at OLI is due to elevated small (<100 nm) particle concentrations, and they mention that these particles are likely too little to act as CCN. As the authors mention, rBC is not necessarily a good indicator of CCN either. Moreover, the range of below-cloud PCASP aerosol concentrations, which are in the size ranges where droplet nucleation would be more likely, were similar between sites (Fig. 4).

Thanks for pointing this out. Please see your comment 17 for a response with respect to HYSPLIT.

We added a new figure (12) to show how PCASP particles can change cloud properties, see above for discussion. We also added a new panel to Figure 3 showing that the increase in CN is particularly related to particles in the 3 to 10 nm range. We also added a section describing how sub-100 nm particles might impact cloud properties to the discussion of Fig. 14:

In addition, some data points lie above the 1:1 line which might indicate that particles smaller than the PCASP size range (i.e. < 100 nm) are acting as

CCN (Leaitch et al., 2016). Further, the assumption that the below-cloud aerosol properties govern the cloud microphysical properties might not be true for all clouds depending on sub-cloud vertical mixing. Therefore, we identified all clouds where the above-cloud PCASP concentration is larger than below-cloud (red dots in Fig 14), and indeed half of these clouds are above the 1:1 line. When using the above-cloud concentration for these clouds, only two of these clouds are above the 1:1 line. However, there are still 11 more clouds above the 1:1 line. Since these clouds generally feature PCASP concentrations < 50 cm-3, the fact that they are above the 1:1 line could be related to increasing sampling errors for small concentrations. They may also confirm the finding of Leaitch et al. (2016) that aerosols below 100 nm can act as CCN for thin clouds.

b) "Reduced reff (between 9 and 12  $\mu$ m) of OLI clouds correspond to increased CN and rBC concentrations" Convincing supporting evidence for the validity and meaningfulness of this trend has not yet been presented (see specific comment #1).

See response to specific comment #1 above.

c) "the mean size of cloud-associated rBC particles is smaller at OLI which is consistent with freshly emitted, less aged particles" There were smaller particles at OLI, and these small particles probably were less aged, but don't the CN and PCASP data indicate that these particles are likely too small to act as efficient CCN?

We removed that argument because it was actually wrong: Because the SP2 does not measure the coating of black carbon particles, smaller particles as measured by the SP2 do not indicate fresher emissions. However, they indicate a different source:

This is consistent with aging of rBC during atmospheric transport, and supports the idea that rBC measurements around OLI are associated with local emissions from Prudhoe Bay and not transported fire emissions. rBC can originate from biomass burning as well as anthropogenic sources, but particle size is smaller for the latter (Schwarz et al., 2008).

In addition, we added that sub-100 nm might also impact cloud properties as discussed above.

d) "clouds [at OLI] were found to be frequently connected to enhanced CN and accumulation mode concentrations" I think the authors meant "associated with" instead of "connected to"? But again, the presence of smaller particles at OLI does not necessarily indicate that they are participating in cloud processes. As I mentioned, the range of below-cloud PCASP aerosol concentrations, which are in the size ranges where droplet nucleation would be more likely, were similar between sites (Fig. 4). As the authors mentioned in the text, particles <100 nm in diameter are likely to be less efficient CCN. In summary, there wasn't convincing evidence that oil-field aerosols did have a discernable impact on cloud properties.

We changed the wording as suggested and removed "connected to".

Please see above for the more general issues raised. We modified the conclusions accordingly:

Finally, while no enhanced concentrations of larger accumulation mode particles were observed for OLI the clouds identified as most likely influenced by anthropogenic activities have significantly higher PCASP concentrations for OLI than for the remaining clouds (Fig. 3), clouds there were found to be frequently connected to enhanced CN and accumulation mode concentrations 12). However, the PCASP concentration of local clouds is not significantly higher than at NSA which might be related to a higher background of PCASP particle concentrations at NSA.

More minor comments (not in order of importance):

3) P2l23: It might be helpful to note upfront that liquid clouds are not that common in the Arctic, compared to ice and mixed phase clouds?

While the reviewer is right that there are more mixed phase and ice clouds year round, low altitude liquid clouds are frequent in summer as temperatures are typically above freezing in Northern Alaska. We added to the introduction of the data set:

During ACME-V, 156 (60%) of the 258 vertically sampled clouds were classified as liquid (see below for thresholds), showing that liquid clouds are frequent during the summer time in Northern Alaska.

4) P5l23: "While Arctic Haze was not observed during ACME-V, transported emissions from forest fires can contribute significantly to summertime aerosol loading in the Arctic?" Earlier Arctic haze was defined as long-range transport of aerosol particles from lower latitudes. Here it seems like the long-range transport of smoke is excluded from consideration as Arctic haze. I think it would help to clarify this sentence, especially since smoke does contribute to what most people define as Arctic haze (e.g., Warneke (2010)). Also, how do you know that what you are referring to here as Arctic haze (Anthropogenic pollution from lower latitudes? Long-range aerosol transport?) was not observed during ACME-V? Is this statement based on the fact that typical precipitation and air mass transports during the summer season make the presence of long-range aerosol transport less likely, or on some extra analysis you did to determine this? Please specify.

We modified that statement, because long range transport was indeed present during ACME-V as found by Creamean et al. (2017), but it will likely not impact our analysis. However, we decided not to use the term 'Arctic Haze', because it is usually related to winter and spring time long range transport. For this, we added a section to the end of

the paragraph and also state that we treat long range transport separately from forest fires.

For ACME-V, Creamean et al. (2017) classified only four flights as impacted by long range transport from lower latitudes not related to forest fires. During these flights, only a single cloud was sampled in the vicinity of OLI or NSA which had one of the lowest aerosol concentrations measured in the whole data set. Therefore we are confident that our analysis is not strongly impacted by this kind of long range transport events.

5) Last paragraph of section 2: This paragraph is a little confusing, and it would help if the authors were more clear about their methods. What was the order and/or priority of the steps to ID long-range smoke transport? From where were the CO data obtained? It says the vertical profiles of rBC and CO were manually inspected ? but what were the criteria relating to those parameters, and what were those criteria specifically used to do? Were the 5 locations where HYSPLIT back trajectories were obtained chosen based on MODIS fire locations?

We extended the paragraph to make the methodology more clear:

Transported emissions from forest fires can contribute significantly to summertime aerosol loading in the Arctic (Law and Stohl, 2007) (Law and Stohl, 2007); (Creamean et al., 2017). Therefore, we manually inspected the vertical profiles of rBC and carbon monoxide (CO) CO, which together constitute a good tracer for biomass burning (Warneke et al., 2009, 2010). Typically, these layers are found aloft (Roiger et al., 2015), allowing us to use vertical profiles obtained by the aircraft to aid in their identification. For each spiral obtained at the two sites, elevated layers with  $CO \ge 0.1$  ppmv or  $rBC \ge 20$ ng  $kg^{-1}$  were flagged as corresponding to forest fires. Local emissions, on the other hand, are expected to be found in a layer connected to the surface. Note that the data impacted by forest fires were only removed for spirals above OLI, NSA, and TOI. For clear-air observations during level flight legs between sites, it is generally impossible to determine whether a layer is connected to the surface or elevated. [...] For the spirals, data identified as originating from forest fire either from manual inspection or according to HYSPLIT, were removed from subsequent analysis.

Regarding MODIS, the five locations chosen were based on the general location of fires detected from MODIS. For the daily HYSPLIT simulations, the five source points were turned on or off, depending on the fire activity in that area for that day, as indicated by MODIS. We modified

 Creamean et al. (2017)) from the Moderate Resolution Imaging Spectroradiometer (MODIS) on the Aqua and Terra satellites obtained using brightness temperature measurements in the 4 and 11  $\mu$ m channels (Giglio et al., 2003; Giglio, 2013).

**6) Fig1: Maybe include the MODIS fire locations?**

This is an excellent suggestion. However, we decided only to show the locations of the fires assumed in HYSPLIT in order to avoid cluttering of the Figure. See Figure 5 of Creamean et al. (2017) for a map of the forest fire locations used.

7) P6l11-12: "The data presented are limited to observations obtained below 500 m in order to demonstrate the impact of local emissions and reduce the impact of forest fires." Up to this point in the text, the reader has not been presented with any strong evidence that smoke (or long-range anthropogenic pollution, for that matter) was not present below 500m. To avoid confusing the reader, please provide this information.

Thank you for pointing this out. We modified the text:

The data presented . As discussed above, removal of data potentially impacted by forest fires is only possible for the spirals. Therefore, the data presented in Fig. 3 are limited to observations obtained below 500 m in order to demonstrate the impact of local emissions and reduce the impact , because transported emissions of forest fires were typically at higher altitudes during ACME-V (for a detailed study of aerosol properties during ACME-V see Creamean et al. (2017)).

8) P6l29-31: It would be helpful to use consistent units here to aid comparisons with the data in this study. Also, the rBC concentrations referenced from Zamora et al., were for background air masses specifically thought to not be influenced by smoke (see their Table 6). From their Figure 7, smoky rBC values ranged somewhere between the order of 1-103 ng/m3 (average appears on the order of 102 ng/m3). Also, the majority of rBC values presented in Figure 3 are much below the median 20-30 ng/kg values from Roiger et al. So what I interpret the rBC from Figure 3 to mean is that the region was very clean with respect to rBC, except near OLI.

We agree with the reviewer and modified the section.

rBC background concentrations appear to be similar to values found by Zamora et al. (2016) (1-16 ng/m3) and Roiger et al. (2015)(median 20-30 ng /kg) for summertime transported forest fire plumes in the Arctic. Other studies (Warneke et al., 2009; Schwarz et al., 2010) found up to one order of magnitude higher rBC concentrations in the Arctic which is more similar to the maximum values we observed around OLI background observations made by Zamora et al. (2016) and Roiger et al. (2015). It should be noted that

emissions related to forest fires led to concentrations as high as 600 to 1000 ng kg-1 during ACME-V (mostly at altitudes above 500 m, Creamean et al., 2017) which were also observed in other data sets (Warneke et al., 2009); (Schwarz et al., 2010); (Zamora et al., 2016). Consequently, the emissions from anthropogenic sources in the OLI region are about a magnitude lower

Regarding the Roiger paper, the 20-30 ng/m3 values correspond to mean values connected to forest fires, i.e. they are not background values. Their Fig 12a, blue dashed line shows their background values are actually very similar (< 5ng/kg) to ours.

Regarding the Zamora paper, we confirmed with the corresponding author of that study that Fig 7 shows data in the range between 2 to 1235 ng C/m3 (and not  $\mu$ g C/m3 as stated in their caption) which is consistent with our findings.

9) P717-13: Your expectation that that there might be more smoke influence at the NSA site would be consistent with literature observations of general smoke particle sizes being larger than the PCASP minimum size range of 100 nm (e.g., Kondo et al. (2011); Sakamoto et al. (2015)). However, I am confused by your statement that, "An alternative explanation could be ... more aerosol processing by precipitation (e.g. Hoppel et al., 1990)." I believe that the Hoppel et al. reference refers to non-precipitating cloud processing of aerosols, correct? So do you mean "processing by clouds" instead of "processing by precipitation "? In which case, why did you mention collision-coalescence and precipitation being the causes for the differences? Or did you mean "removal" or "scavenging" here instead of "processing" (in which case why the subsequent sentence regarding aerosol in-cloud processing)? Other readers might be confused as well, so clarification here would be helpful.

We added the suggested references

Transported forest fire aerosols are often larger than the PCASP detection threshold of 100 nm as shown by Kondo et al. (2011) and Sakamoto et al. (2015).

Regarding the study by Hoppel, the reviewer right that we picked the wrong reference by accident. We replaced the reference with Feingold et al. (1996)

10) P7l18: "While mean particle size generally increases with decreasing CN concentration, ?" Looking that figure 4, this is not clear to me. I suggest taking this statement out, since it doesn't seem that necessary. Or, if you disagree, you could try plotting on a log-scale to see if the pattern emerges more clearly or switching the x and z axes to show the trend better?

Because it did not belong to the main focus of the article, we removed the figure and the corresponding statement.

11) P7l19-20: "the variability of PCASP mean size is rather low for CN concentrations > 600 cm-3, which is consistent with the idea that particles have already experienced growth" Are there other things that might cause the same trend? Might be worth mentioning.

Same as above: we removed the figure and the statement.

12) Fig.4: why was the y-axis cut off at 4000? To provide aerosol context for the cloud data? To provide better context for the study as a whole, you might consider plotting on a log scale to include all the data, or at least mentioning in the caption that a fraction of the data are excluded for whatever your reason.

Same as above: we removed the figure and the statement.

13) P9l6-7: "It is not possible to study the cloud life cycle using aircraft in-situ observations, but the potential for impact on cloud life cycle can be estimated by?" I suggest moving your qualifiers in the last paragraph of this section up here so that you can better clarify upfront that what you are estimating is a factor important to cloud lifetime, but your estimate is not necessarily indicative of cloud lifetime itself.

We followed the reviewer's suggestion and changed the order.

14) P10l4-5: "This is caused by reduced broadening of the drop size distribution towards large drops at OLI (not shown), consistent with cloud chamber experiments (Gunn and Phillips, 1957)." Maybe I am not understanding something, but shouldn't that kind of broadening be apparent in Fig. 5? Why not mention that here? Also, what specifically is consistent with this cloud chamber experiment?

Figure 5 (now 4) shows only the effective radius which does not provide any information about the breadth of the drop size distribution. Regarding the cloud chamber experiment, we decided to keep the reference for historical context, but extended:

This is caused by reduced broadening of the drop size distribution towards large drops at OLI ( not shown Fig. 5.b ), consistent with cloud chamber experiments (Gunn and Phillips, 1957) the experiments of Gunn and Phillips (1957), who produced similar results when ingesting polluted background air into their cloud chamber.

Please see the new Figure 5.b for information on the breadth of the distribution.

15) P10l9: "typical reff values are reduced at OLI in comparison to NSA for the same LWC." I don't see that consistently in Fig. 5. What about at high LWCs, for example? Did you mean within a specific LWC range? If so, please specify.

We added a new Figure 5a to highlight the relevant LWC range and to apply a statistical test:

Fig. 5.a reveals that this difference is statistically significant for most LWC  $> 0.1 \text{ g m}^{-3}$  according to a Welch's t-test (Welch, 1947) with a 5% confidence interval (Note that a 5% confidence interval is always used in this study unless noted otherwise).

16) P10l13: more info on this droplet fall velocity parameterization would be helpful.

A detailed description would be out of the scope of the paper and parallel the original publication. However, we added:

... which provides a continuous solution over the entire drop size range in dependence of the Best and Reynolds number

17) P1116: "For OLI (NSA), 62% (16%) of all ACME-V cloud observations can be traced back to surface emissions originating from the Prudhoe Bay oilfields." I am confused here. How did were these values calculated? What does it mean that a certain fraction of cloud observations were traced back to surface emissions? That some fraction of the aerosol particles in the air masses containing the clouds were likely to have originated from the oilfields? Please clarify in the text.

Thanks for pointing this out. We clarified:

For OLI (NSA), 62 50 % (16%) of all data points observed within clouds during ACME-V <del>cloud observations</del> can be traced back to surface emissions (i.e. mass concentration > 0 according to HYSPLIT) originating from the Prudhoe Bay oilfields.

Note that the reduction from 62 to 50% is due to the exclusion of extremely low clouds at OLI as discussed above.

18) Fig. 7 and text citing it: It is not clear to me how one can get the mass concentration at a certain location unless the emissions at the point source are known. Is this a relative mass concentration? Was emissions information available? If so, please provide it. Also, what are the errors in the co-location of the modeled plume and the actual plume? Please discuss. If you don't have this information, I recommend removing this figure and just stating in the text that HYSPLIT back trajectories indicate a higher influence of oil field aerosols at OLI than at NSA x% of the time, and that one would expect higher oil field aerosol concentrations at OLI than at NSA due to proximity of the source. Figs. 8 and 9 probably provide more trustworthy information on aerosol concentrations anyways (please correct me if I am wrong).

Because HYSPLIT provides a relative unit (mass per volume), we do not have to know the emission rate. We added to the end of the paragraph:

Note that HYSPLIT provides only a relative emission rate in 'mass  $m^{-3}$ '

because the actual emission rate in the Prudhoe Bay region is unknown.

We decided to keep the Figure, because it also shows the variability for similar reff and LWC due to the smaller bins. We agree that HYSPLIT uncertainties might be high:

Note that given the coarse resolution of the forcing model (1°) and the complexity of the Arctic boundary layer, HYSPLIT is used here only in a qualitative way, and not to select locally impacted clouds.

19) P1618-9: Regarding the statement that, "As a consequence, the breadth of the size distribution of liquid droplets has to be smaller at OLI", why not just provide the size distribution data from the campaign?

The size distribution depends strongly on the LWC. Therefore we decided to show the mean standard deviation of droplet size as a function of LWC in the new Figure 5.b.

20) p. 17/10-12: "While forest fire cases have typically higher aerosol concentrations and consequently droplet concentrations, their inclusion into the estimation of ACI does not substantially alter the found relationship." I disagree with this statement. Based on the arguments from specific comment #2 above, the authors admission that they might not have been able to completely exclude smoke influenced air masses from the study, and the similar ACI values for smoke influenced cases, for all we know, smoke was driving the relationships? or did I miss something?

We are not sure whether we understand the reviewer here properly. We updated Figure 14 to show that clouds potentially impacted by local emissions are between forest fires and the remaining clouds with respect to aerosol concentration. However, the sample size is too small to estimate ACI for them separately. We added:

As already discussed, clouds associated with local emissions have lower PCASP (and likely accumulation mode) concentrations than forest fires, but still have larger concentrations than for the other clouds.

We added to the end of the section that the goal is only to investigate whether there are differences in the mechanism how aerosols change cloud properties.:

Therefore, we conclude that a significant difference of difference in ACI between local emissions and forest fires cannot be found, given the limited data set. This refers only to the mechanisms through which aerosols change cloud properties, and does not imply that local emissions do not change cloud properties.

We also added the standard error of ACI obtained by the fit in order to show that the uncertainty is larger than the difference.

Technical corrections:

P3l15: You cite a 1993 paper for 2015 observations. Please clarify what you meant by citing this paper here.

This is not a paper, but a data set. We modified the reference to make this more clear.

P6l15: Just a suggestion, get rid of "For the SP2", since people may not remember what SP2 is, and it is not necessary to the sentence.

Changed to from the SP2 probe.

P6l20: Suggest combining this sentence with the following sentence to make it clear what specific pattern(s) is/are being deemed similar (the skewness and spatial distribution as opposed to the median?).

We modified the sentence to indicate it is about spatial patterns:

CN measurements from the CPC show a similar pattern spatial pattern similar to the SP2 even though the increased values are distributed over a larger area

P612-4: The units in this sentence are confusing. Also, can you clarify in the manuscript that the latter two numbers are for wet deposition scavenging (or some other kind of scavenging)?

We agree that the units are confusing, but they are in accordance with the HYSPLIT manual. The numbers are indeed for wet deposition scavenging, we added this information to the text.

P811: suggest rewrite to "As mentioned above, clouds known to be impacted by forest fires have been removed" to better reflect the uncertainty you discussed in the previous section.

Thanks for the suggestion, changed.

Fig. 5: please describe in the caption what the dashed line indicates. Also suggest adding in the term "NSA", since that is what you talk about in the text referring to this figure. Also, I found the label "number of observations [%]" to be very confusing. I recommend changing this to something like, "% of total observations."

Good suggestions, we modified the figures and the caption accordingly.

Fig.6: The caption says OLI-NSA, but the figure says NSA-OLI? Also, can you make the axes consistent between Fig. 5 and 6? That would help enable comparisons between the two. Lastly, in the caption it says that, "The green dots highlight data points with less than five observations." Please clarify in the caption whether those 5 observations include points at both sites added together, or at each site individually. Thanks for noting the wrong caption and varying axes, we fixed that. The dots indicated less that 5 observations for one of the sites. However, we use the dots now to indicate whether the difference is significant or not.

Suggest using site abbreviations throughout the paper and in the figures after they have been defined, instead of using a mix of the site names and the abbreviations to avoid confusion.

Excellent suggestions, we modified the figures.

P1215: Please clarify: what "this" do you refer to when you say, "This means?"?

We replaced 'This means that' with 'As a consequence'

Fig 12: concentration not cooncentration on the x-axis.

P17l19: from not form.

Both fixed, thanks.

**2 Reviewer III**

The authors use airborne observations from June-September, 2015 within 90 km of two DOE-ARM sites (North Slope Alaska or NSA and Olitok Point or OLI) to study potential effects of changes in aerosol particles on Arctic liquid water clouds. The aerosol observations are limited to physical measurements with a PCASP (particles larger than about 100 nm) and a condensation particle counter (CPC; particles large than about 3 nm as well as measurements of black carbon (rBC). The main objective is to see if clouds formed on particles from anthropogenic activities in the OLI area result in differences in cloud microphysics compared with cloud formed on particles observed in the NSA area. The topic is relevant for ACP, the paper is well organized and interesting, and I believe the overall results are useful. However, the paper is not currently ready for publication.

Major comments:

1) The paper leads up to section 6 (Quantification of cloud aerosol interaction) by accumulating information suggesting the microphysics of the OLI clouds are impacted by local emissions. That notion is then set aside in section 6 based on the calculated ACI indices. However, the ACI calculation in inappropriate for these observations if particles smaller than 100 nm are nucleating droplets. Drawing a 1:1 line in Figure 12 indicates some points above. Assuming the measurements are reasonably accurate, which the authors do not discuss, then relatively few droplets are nucleated on particles smaller than the lower limit of the PCASP. The larger deviations above the 1:1 line are towards lower aerosol concentrations, which would be consistent with Leaitch et al. (ACP, 2016) if there are sufficient particles smaller than 100 nm to explain the deviations. As it stands, the ACI discussion tells us only that there is some impact of PCASP-sized particles on the NSA cloud observations, which has already been mentioned in connection with Figure 10 and is not the focus of the paper. The fundamental result could be more clearly shown using Figure 12 with straight concentrations rather than natural logarithms. What do Figures 8, 9, 10 and 11 tell us that cannot be found from a modified Figure 12?

It was not our intent to set the notion "local emission impact cloud properties" aside with Fig. 12 (now 14) and we think that this is not supported by the measurements. The intent of the ACI discussion is to investigate how the mechanisms connecting PCASP concentrations to clouds are different when connected to local and forest fire emissions. To make this more clear, we added

Therefore, we conclude that a significant difference of difference in ACI between local emissions and forest fires cannot be found, given the limited data set. This refers only to the mechanisms through which aerosols change cloud properties, and does not imply that local emissions do not change cloud properties.

We agree that the argument on how local emissions change cloud properties was not convincing. Therefore, we identified clouds potentially impacted by local emissions and show that they have larger aerosol concentrations at OLI than the remaining clouds (Fig 14). The new Figure 12 discusses that in greater detail. Moreover, we agree that sub-100 nm particles could have impacted the analysis:

In addition, some data points lie above the 1:1 line which might indicate that particles smaller than the PCASP size range (i.e. < 100 nm) are acting as CCN (Leaitch et al., 2016). Further, the assumption that the below-cloud aerosol properties govern the cloud microphysical properties might not be true for all clouds depending on sub-cloud vertical mixing. Therefore, we identified all clouds where the above-cloud PCASP concentration is larger than below-cloud (red dots in Fig 14), and indeed half of these clouds are above the 1:1 line. When using the above-cloud concentration for these clouds, only two of these clouds are above the 1:1 line. However, there are still 11 more clouds above the 1:1 line. Since these clouds generally feature PCASP concentrations < 50 cm-3, the fact that they are above the 1:1 line could be related to increasing sampling errors for small concentrations. They may also confirm the finding of Leaitch et al. (2016) that aerosols below 100 nm can act as CCN for thin clouds.

2) Concerning Figure 6 and related discussion on page 10, I have the following questions and remarks:

a) The differential collection growth rates (OLI minus NSA) range between 0.1 and 1

(units of kg/sm3). The mean collection growth rates vary from less than 1e-12 to about 5e-6 with the same units. How can the differential rates be higher than the mean rates? Are the differential values ratios (e.g. OLI-NSA/NSA) rather than absolute values?

Thanks for pointing this out, the difference is actually in log scale, i.e. the plot shows actually a ratio. We updated figure and text accordingly.

b) On lines 3-4, you say that C is lower at OLI compared with NSA for constant LWC and Reff. This is very difficult to see in 6a. Regardless of whether the differentials (OLI-NSA) are absolute or ratios, they are higher, not lower. Also, how do I look at constant LWC and Reff in these plots?

Sorry, we missed to update the Figure caption as we swapped OLI and NSA last minute, it is NSA/OLI. The values are larger for NSA as stated in the text correctly. We changed the caption accordingly. The axes of the plot are LWC and reff, therefore every pixel of the plot is for constant reff and LWC (or at least for a small interval).

c) On lines 10 and 11, you say that the mean value of C (averaged for Reff) is 1 to 1.5 orders of magnitude reduced at OLI for a constant LWC. Yet the mean OLI growth curve lies to the right of the NSA curve, which indicates a higher mean collection growth rate.

We apologize, when doing the last minute swap of the sites, we also overlooked to update the legend. The description in the text is correct, we updated the figure legend.

d) The same apparent discrepancies are present for the rainfall rates in Fig. 6b.

Same as above.

3) The aerosol observations are restricted to below 500 m-msl. Please indicate the altitude range for the cloud observations. Please indicate how you know that the aerosol below cloud was connected with the cloud above and not isolated by temperature inversions, which can happen in the relatively stable environment of the Arctic.

This is a misunderstanding, the 500 m limit is only applied to the aerosol observations presented in Figure 3 in order to avoid contamination by forest fires. We use only cloud observations in the vicinity of the sites where vertical profiles were obtained. Therefore we can remove forest fires by looking at elevated layers. For the legs between the sites we cannot do that because we do not know whether an observed aerosol layer is elevated or not. Therefore we applied a 500 m threshold instead. For Figs 8-14, the aerosol concentration is obtained directly under the cloud, whether the cloud base is below or above 500 m does not matter. See also Figs. 2 and 6 in Creamean et al 2017. We updated the description of Fig 3 accordingly:

The data presented . As discussed above, removal of data potentially

impacted by forest fires is only possible for the spirals. Therefore, the data presented in Fig. 3 are limited to observations obtained below 500 m in order to demonstrate the impact of local emissions and reduce the impact , because transported emissions of forest fires were typically at higher altitudes during ACME-V (for a detailed study of aerosol properties during ACME-V see Creamean et al., 2017)

Regarding the range of cloud observations, we added:

Cloud base varied between 178 m and 5346 m with a median of 1498 m.

Minor comments:

4) Page 2, lines 4-5 - Aerosol number concentration or mass concentration? If number, what sizes? This statement is very simplistic.

We clarified "cloud condensation nuclei (CCN) concentration"

5) Page 3, line 18 - what are 'bulk' probes?

Bulk probes measure bulk cloud properties such as LWC. We removed that statement because bulk probes are not used in this study.

6) Pages 3 and 4 - If not described, references are needed for how the CDP, DCDP and OAPs were evaluated and calibrated during the study.

We added:

For the 2DS probe, the evaluation of particle sizing and sample area determination was done following Korolev et al. (1991). The sample areas of CDP and FCDP were determined by their manufacturer using the technique described by Lance et al. (2010). The droplet size response for CDP and FCDP was calibrated weekly using glass beads in field. In addition, LWC was measured by a multi-element water content system (WCM-200) and used for evaluating the in-flight performance of the 2DS, CDP and FCDP (King et al., 1978, 1981, 1985).

7) Page 4, line 6 - A droplet threshold of 10/cc may not be appropriate for Arctic summer clouds (e.g. Leaitch et al., ACP, 2016), and it does not allow consideration of situations such as discussed by Mauritsen et al. (ACP, 2011). Please discuss.

Thank you for this comment. We tested also the use of lower thresholds. For  $5 \times 10^6$  m-3, the number of clouds increased only by two and they were classified as associated with forest fires. For even lower thresholds, we found artifacts in the cloud classifications. In addition, we see a change in cloud properties mostly for LWC > of 0.1 g/m3, making us confident that clouds with very small LWC are not relevant for our study.

However, we agree, that a discussion about the threshold is necessary, so we added:

We also evaluated the use of a lower cloud threshold (5 cm-3 in accordance with Hobbs and Rangno (1998)), but this only increased the number of observed clouds by two, both of which were classified as impacted by forest fire (see below). In order to avoid sampling errors due to small sample sizes, we use the larger threshold (10 cm-3) in the following.

8) Page 4, line 10 - what are 'tiny' particles?

Please see the line above in the manuscript:

The habit classification scheme differentiates between spherical particles, tiny particles which are too small to be classified and various forms of ice crystals.

9) Page 5 - Describe the inlet for particles measured with the CPCs.

We added:

Aerosol particles were sampled through an isokinetic inlet with an upper size cut of  $5 \ \mu m$  (Zaveri et al., 2010; Dolgos and Martins, 2014)

10) Page 5, line 11 - How was the PCASP calibrated during the study?

We added:

The PCASP was calibrated using both size selected ammonium sulphate particles and monodisperse polystyrene latex (PSL) spheres. The sizing accuracy was checked weekly in the field using PSL particles following Cai et al. (2013).

Is the lower detection limit truly 100 nm (e.g. Liu et al.: Response of Particle Measuring Systems airborne ASASP and PCASP to NaCl and latex particles, Aerosol Sci. Technol., 16, 83-95, 1992)?

The lower detection limit of the PCASP used is 90 nm, but we omitted the 90-100 nm bin due to the low counting efficiency.

11) Page 5, line 20 - How as the SP2 calibrated?

We added:

The applied SP2 calibration methods using ambient BC and fullerene soot are described in detail by Gysel et al. (2011) and Irwin et al. (2013). The fullerene soot and PSL calibration were performed twice during this field campaign and the sensitivity of the SP2 was found to be stable to around 10% for fullerene soot particles resulting in an estimated SP2 measurement uncertainty of 10%.

12) Page 5, line 23 - While Arctic Haze is not common during the summer, how can you be certain it was not observed? PCASP number concentrations of 150-200/cc may be representative of Arctic Haze (e.g. Leaitch et al., J. Atmos. Chem, 9, 187-211, 1989).

The reviewer is right, long range transport was actually found by Creamean et al. (2017) for AMCE-V. We decided not to use the term 'Arctic Haze', because it is usually related to winter and spring time long range transport. However, this does not impact our cloud analysis, we updated:

For ACME-V, Creamean et al. (2017) classified only four flights as impacted by long range transport from lower latitudes not related to forest fires. During these flights, only a single cloud was sampled in the vicinity of OLI or NSA which had one of the lowest aerosol concentrations measured in the whole data set. Therefore we are confident that our analysis is not strongly impacted by this kind of long range transport events.

13) Page 5, line 35 - A particle density of 6 g/cm3 is very high. The density of submicron particles is usually less than 2.5 g/cm3. Please explain.

As stated, this was a HYSPLIT default parameter when deposition is turned on. Additionally, we use the HYSPLIT results only qualitatively, thus, the deposition locations would not change with changing particle density.

14) Page 6, line 12 - aerosol data

Added.

15) Page 6, line 23 - reference for size of freshly emitted soot?

Thank you for asking for a reference, because the reference actually states 15 nm. We updated:

Freshly emitted soot has been shown to be larger than this (> 20 nm), so this range is 15 nm (Zhang et al., 2008), so particles in the 3 to 10 nm size range are likely due to in situ nucleation of aerosol particles from gas phase precursors (i.e., formation of new particles as compared to secondary aerosol formation, where gases condense onto preexisting aerosol, Kulmala et al., 2012)

16) Page 6, line 26 - qualify 'quickly' by assuming sufficient gaseous precusors.

We extended:

Even though the CN dominating the CPC observations are likely too small to act as a— CCN, these small particles can grow to accumulation mode quickly given sufficient gaseous precursors, potentially creating a particle population capable of acting as CCN (Jaenicke, 1980).

17) Page 7, line 22 - smallER

Thanks, fixed.

18) Page 7 - In Fig. 4, there are interesting similarities between the small group of yellow points associated with each site. Both groups show increases in PCASP size with increasing CPC. This curious group is related to cloud for the OLI case but not the NSA case. Can you identify a connection?

Because it did not belong to the main focus of the article, we removed the figure.

19) Page 9, line 1 - Indicate the reason for the 16 um line in the caption of Fig. 5. Also, in Fig. 5, please add a line showing how LWC vs LER varies assuming the mean droplet number concentration for each location. What are those mean droplet concentrations?

We added a description of the dashed line to the caption of Figure 5 (now 4). We added a new Figure 5 to show how mean reff and the standard deviation of the drop size distribution changes as a function of LWC. We added:

Fig. 5.a reveals that this difference is statistically significant for most LWC  $> 0.1 \text{ g m}^{-3}$  according to a Welch's t-test (Welch, 1947) with a 5% confidence interval (Note that a 5% confidence interval is always used in this study unless noted otherwise). [...] For TOI, the mean  $r_{eff}$  is significantly different than those at NSA for LWC  $> 0.02 \text{ g m}^{-3}$ . Fig. 5.b reveals that not only the mean  $r_{eff}$  is reduced at OLI, but also the breadth of the distribution as shown by the standard deviation. This difference is significant for most data points with LWC  $> 0.1 \text{ g m}^{-3}$ .

20) Page 12, lines 1-3 - Is there evidence that OLI emissions impacted any of the NSA observations?

Indeed, this cannot be excluded because both our Hysplit simulations and Kolesar et al. (2017) found found that emissions can make it to NSA. Therefore we extended:

The HYSPLIT simulations (Fig. 7) show that the mass concentration originating from local pollution sources is on average more than two order of magnitudes can be a substantially higher at OLI than at NSA. These simulations indicate that relative to NSA, the number of clouds impacted by local emission is increased higher at OLI and these clouds are impacted by a larger amount of aerosol particles by mass. However, an impact of local emissions on cloud properties is also possible at NSA, although less frequently than at OLI.

We added to the conclusions:

Given the limited data set, we found ten of 24 clouds at OLI, but only one of 16 clouds at NSA which might be impacted by anthropogenic emissions.

21) Page 12, lines 9-10 - Reduced Reff with increased rBC is not so clear; these plots have a qualitative aspect to them. At the higher LWC (> 0.1 g/m3) that may be true, but below 0.1 it appears that the opposite may be true.

Because we decreased the pixel size, this effect is now visible for all LWC, albeit not all clouds are impacted

22) Page 12, lines 15-16 - When averaged over a large number of observations. Also, the "notion" is commonly anticipated for clouds with higher LWC (roughly >0.1 g/m3; e.g. Leaitch et al., JGR, 1992) when effects of evaporation, dissipation and precipitation are reduced factors.

The updated Figure shows smaller bins, i.e. there is less averaging. We reworded that statement:

However, for OLI there are more enhanced rBC concentrations (> 4 ng kg-1) for intermediate  $r_{eff}$ -values (5-10  $\mu$ m) consistent with Fig. 3.a. It is interesting to note that this intermediate region is consistent with enhanced local particle concentrations according to HYSPLIT.

We also added the proposed reference.

23) Page 12, Lines 18-20 - The use of monotonic is not justified here.

True, we removed the word monotonic.

24) Page 13, lines 8-9 - How many CCN are needed for cloud formation?

We removed Fig. 4 and the corresponding discussion in order to shorten the manuscript and because it did not contribute to the main focus of the article.

25) Page 15, header for section 6 - You are not discussing an 'interaction' here, only a potential impact of the aerosol on the cloud.

We decided to stick to the term "aerosol cloud interaction" to be consistent with existing literature (e.g. Coopman et al., 2016; Zamora et al., 2016). In addition, clouds do have an impact on aerosols, e.g. by processing and through dynamical feed-backs. Therefore the term interaction is appropriate.

**The observed influence of local anthropogenic pollution on northern Alaskan cloud properties**

Maximilian Maahn1,2, Gijs de Boer1,2, Jessie M. Creamean1,2, Graham Feingold2, Greg M. McFarquhar3, Wei Wu4,5, and Fan Mei6

[revised manuscript text omitted]

---

## Author Response (AR2)

**The observed influence of local anthropogenic pollution on northern Alaskan cloud properties**

**2nd response to reviewers**

M. Maahn, G. de Boer, J. Creamean, G. Feingold, G. McFarquhar, W. Wu, and F. Mei

October 19, 2017

Original Referee comments are in italic

manuscript text is indented, with added text underlined and <del>removed text</del> erossed out.

We would like to thank the reviewers for their second round of detailed and helpful comments. We revised the manuscript and responded to all of the reviewers' comments. Please refer also to the manuscript with all changes highlighted.

**1 Reviewer II**

The authors spent a great deal of effort addressing the reviewer suggestions, which is much appreciated, and the paper is substantially improved. The logic of the paper and the figures are now much clearer, and I am happy to recommend it for publication, after a few minor comments detailed below are addressed.

Specific comments:

p.16, l.20: "Because enhanced rBC and CN concentrations are expected to be good indicators of anthropogenic activity, they are used to isolate clouds impacted by anthropogenic emissions." I agree that rBC is a good indicator, but I am not so sure about CN. In this paper, CN is defined as the number count of aerosols between 3 or 10 nm up to 3 microns in diameter (p.5, l.35). Many local non-combustion derived aerosols pop up in the lower diameter range between 3-60 nm (e.g., Tunved et al. 2013). These very small background particles can be so numerous that they can dominate the CN count (e.g., Zamora et al., 2016). That could be one explanation of the large discrepancy between Figures 8 and 10 in the aerosol distributions relative to reff, why Figure 12 and Figure 8 are so similar, and why there are more brown points in Figure 13b than in Figures 13a and 13c.

Figure 3.b shows that there is an increased concentration of CN in the OLI, region, Fig. 3.c shows that this is also related to the 3 - 10 nm size range, i.e. the CN concentration is strongly controlled by the smallest particles as stated by the reviewer. This can be also seen by the fact that PCASP concentrations are typically one order of magnitude smaller than CPC CN concentrations which is in agreement with the reviewers statement that small particles can dominate the CN count. We wrote that the enhanced CPC concentrations are likely related to the petroleum facilities:

...particles in the 3 to 10 nm size range are likely due to in situ nucleation of aerosol particles from gas phase precursors (i.e., formation of new particles as compared to secondary aerosol formation, where gases condense onto preexisting aerosol, Kulmala et al., 2012). Nucleated aerosols typically have sizes below 3 nm, but quickly grow via condensation and coagulation to sizes > 3 nm (Colbeck and Lazaridis, 2014). This source of nucleated aerosol particles from petroleum and gas extraction activities (e.g., flaring and venting of gas) has been reported by Kolesar et al. (2017) for emissions transported from OLI to NSA.

Because this differential measurement using to instruments might have large uncertainties, we did not use that quantity for cloud classification. However, when using the differential CPC measurement for cloud classification, all but one clouds are classified the very same. But we agree with the reviewer that there are likely also other sources and this has to be stated and added:

Because enhanced rBC and CN concentrations are expected to be good indicators of anthropogenic activity, they are used enhanced in the OLI region (Fig. 3) which is probably related to anthropogenic combustion processes and gas flaring/venting, respectively. Therefore we used these quantities as indicators to isolate clouds impacted by anthropogenic emissions even though there also exist other local sources of small particles (Tunved et al., 2013).

p.17 l.4: "We conclude that CN and rBC particles, which were used to classify local clouds, have the potential to grow to accumulation mode particles measured by the PCASP." Again, I feel that CN is inappropriate to use for this purpose – PCASP con-

**centrations would likely be better.**

The issue with using the PCASP concentrations for cloud classification was that the background concentration was higher at NSA and the anthropogenic signal was small. Therefore, we developed the classification to show the impact of local emissions for OLI clouds.

Also, from the author's definition of CN, the CN data should already include the data in the PCASP ranges. This is confusing to me.

The reviewer is correct that the size ranges of CN and PCASP are overlapping. However, there are much more smaller than larger particles. Therefore, the CPC measurements are governed by particles too small for the PCASP, as can be seen from the large difference between CPC and PCASP.

P.19, l.7: "When applying the linear regression to the data sets corresponding to the two sites separately, the obtained ACI values differ (Table 1), with OLI having a lower ACI value  $(0.12\pm0.05)$  than NSA  $(0.20\pm0.07)$ . Given the small sample size (24 and 16 cases for OLI and NSA) which was caused by the PCASP data being quality-flagged for some cases, it is not possible to determine whether this is caused by a difference in nucleation efficiency between aerosols at the two sites or a random effect." How were the errors in ACI value calculated? Is the difference in ACI values between the two sites significant? Given the error ranges listed here, and the high spread of data shown in Figure 14, it looks likely to me that they are likely not significantly different, in which case discussion of the differences is not appropriate.

That was exactly the point we wanted to make. We clarified:

Given the small sample size (24 and 16 cases for OLI and NSA) which was , caused by the PCASP data being quality-flagged for some cases) and the overlap of the uncertainty ranges (obtained from the linear regression), it is not possible to determine whether this is caused by there is a difference in nucleation efficiency between aerosols at the two sitesor a random effect.

P20, l.5: For the following reasons, I feel that the following new text is too speculative, and suggest removing it: "The lower R2 value for OLI (0.24) in comparison to NSA (0.40) could indicate that the assumption that PCASP particle concentrations represent a good approximation for CCN concentrations is partly violated at OLI." As mentioned, I'd like to know if the ACI values actually were significantly different.

As clarified above, the values are not significantly different. However, we think that the information contained in  $\mathbb{R}^2$  is separate from that because it describes the spread around the regression line.

"This could result from those particles being less aged and consequently less coated by

sulphates and organics in comparison to those observed around NSA." These data are not provided.

We agree that this is not supported by the available observations and decided to say that more explicitly:

This could result from is consistent with those particles being less aged and consequently less coated by sulphates and organics in comparison to those observed around NSA, though detailed observations of chemical composition were not available for this campaign.

"In addition, some data points lie above the 1:1 line which might indicate that particles smaller than the PCASP size range (i.e. ; 100 nm) are acting as CCN (Leaitch et al., 2016)." What figure is being referred to here - Figure 14?

Correct, we added that.

"Further, the assumption that the below-cloud aerosol properties govern the cloud microphysical properties might not be true for all clouds depending on sub-cloud vertical mixing. Therefore, we identified all clouds where the above-cloud PCASP concentration is larger than below-cloud (red dots in Fig 14), and indeed half of these clouds are above the 1:1 line. When using the above-cloud concentration for these clouds, only two of these clouds are above the 1:1 line. However, there are still 11 more clouds above the 1:1 line. Since these clouds generally feature PCASP concentrations < 50 cm-3, the fact that they are above the 1:1 line could be related to increasing sampling errors for small concentrations. They may also confirm the finding of Leaitch et al. (2016) that aerosols below 100 nm can act as CCN for thin clouds." This last argument is based on a very low sample number. Especially due to this low sample size, detailed description of the instrument/sampling error is key to giving any credence to the hypothesis. To "confirm" the Leaitch et al findings, the authors would need to show a lot more information on this error than what is provided here. Without that information, I believe the above discussion is much too speculative.

We would like to keep this argument to present all possible options to the reader. But we agree that the word 'confirm' was misleading and rephrased the sentence to make it more general:

However, there are still 11 more clouds above the 1:1 line. Since these clouds generally feature PCASP concentrations  $< 50 \text{ cm}^{-3}$ , the fact that they are above the 1:1 line could be related to increasing sampling errors for small concentrations. They may also confirm the finding of Leaitch et al. (2016) that , but might be also related to activation of aerosols below 100 nm can act as CCN for thin clouds (Leaitch et al., 2016).

It should also be noted that small sample sizes are unfortunately common when investigating in situ cloud observations. E.g. Leaitch et al. (2016) used only 62 clouds, even 5 less than presented in this study. However, a single cloud data point represents always an average over a certain flight time, thus reducing the error.

Technical suggestions

p.2, l.3: Because of these observations...

Unfortunately, we do not understand to what sentence the reviewer is referring to.

p.5, l.5: Suggest: In the following work

Changed to "in our study".

p.6, l.15: I still disagree with the phrasing here, because while these tracers are emitted by fires, they are also affected by other processes. BC and CO are only good tracers of smoke for in situ data if the other processes affecting the BC/CO concentrations are unimportant at a specific place and time. I suggest rephrasing to something like, "Therefore, we manually inspected the vertical profiles of rBC and CO, which together can be used to trace biomass burning in otherwise clean environments (Warneke et al., 2009, 2010)." (Note: if you make this change, the reference is probably not appropriate anymore, since I believe Warneke et al. used a model to identify smoke-related BC in non-clean environments)

We modified the sentence as suggested and added another reference to Zamora et al. (2016)

For the same reason, on p.6, l.17, I suggest you change to: "For each spiral obtained at the two sites, elevated layers with  $CO \ge 0.1$  ppmv or  $rBC \ge 20$  ng kg-1 were flagged as potentially corresponding to forest fires."

We added 'potentially' as suggested.

p. 6, l.18-19: "Local emissions, on the other hand, are expected to be found in a layer connected to the surface" – how sure are you about that? Do you mean "Local emissions, on the other hand, are expected to be concentrated in the layer..."?

We expect the concentrations to be higher in the boundary layer, because they would be more disperse at higher altitudes. We rephrased the sentence

Local emissions, on the other hand, are expected to be found in a layer connected to the surface concentrated in the boundary layer.

p.8, l. 29: e.g.?

This was a broken reference to Feingold et al. (1996). The reference was broken only in

the version with all modifications highlighted.

p.10, l.24: what does the question mark mean?

Same as above (Leaitch et al., 1992).

Figure 1: it would be helpful to state in the caption how fire source was identified

We added "based on MODIS thermal anomaly observations".

Good point, added.

**The observed influence of local anthropogenic pollution on northern Alaskan cloud properties**

Maximilian Maahn1,2, Gijs de Boer1,2, Jessie M. Creamean1,2, Graham Feingold2, Greg M. McFarquhar3, Wei Wu4,5, and Fan Mei6

[revised manuscript text omitted]